# PAMDP: Interact to Persona Alignment via a Partially Observable Markov Decision Process

**Zhe Yang [1], Yi Huang [1,2] \*, Si Chen [1], Xiaoting Wu [1], Jingyu Yao [1], Junlan Feng [1] \***
[1] JIUTIAN Research; [2] Department of Computer Science and Technology, Tsinghua University, China
{yangzhe, huangyi, chensi, wuxiaoting, yaojingyu, fengjunlan}@cmjt.chinamobile.com

## Abstract

The interaction process of comprehending user-specific nuances and adapting to their preferences represents a pivotal consideration for Persona Large Language Models, as it more authentically mirrors genuine dialogue dynamics than adherence to general human value alignment. In this paper, we conceptualize this "Interact to **P**ersona **A**lignment" challenge as a Partially Observable **M**arkov **D**ecision **P**rocess, abbreviated as Persona Alignment MDP (PAMDP), wherein the user's dynamically evolving profile through interaction is treated as an unobservable variable to the assistant. Grounded in this formulation, we propose a dual-critic reinforcement learning framework, with a continuous latent space action representing the assistant's utterance. We evaluate our approach on both offline datasets and the online simulator, ultimately demonstrating its effectiveness.

## 1 Introduction

The alignment of large language models (LLMs) with human value preferences is typically achieved through post-training alignment techniques, particularly supervised fine-tuning (SFT) and reinforcement learning from human feedback (RLHF) Ouyang et al. (2022), are commonly implemented as standard approaches in the field. These methods aim to optimize models to satisfy general human preferences, including criteria such as helpfulness, harmlessness, and honesty. However, human preferences exhibit significant heterogeneity across different user groups and individuals. Moreover, even for a single user, preferences may demonstrate subtle variations depending on contextual factors. The single reward model in RLHF overlooks the rich diversity of human preferences inherent in data collected from multiple users, which leads to the inability of LLMs to align with user-specific preferences Wu et al. (2023). As Chakraborty et al. theoretically and empirically demonstrates, the conventional RLHF approach with a monolithic reward mechanism cannot adequately capture the full spectrum of diverse human preferences. This limitation underscores the need for more nuanced alignment frameworks capable of accommodating preference variability.

To enhance personalization, user-specific data—such as profiles, interaction histories, and behavioral patterns—are utilized to generate responses that align with individual preferences and contextual needs Liu et al. (2025); Li et al. (2025). This adaptive approach ensures greater relevance and customization in model outputs. However, obtaining sufficient user-specific data for personalization remains nontrivial due to privacy restrictions and the inherent sensitivity of personal behavioral data. Wu et al. propose training LLMs to align with individual preferences through interactions. To achieve this goal, they establish distinct user personas and multi-turn preference data to fine-tune LLMs, enabling it to explicitly infer user preferences during interaction and generate personalized responses. As the conversation goes deeper, the alignment level is expected to improve iteratively with each conversational turn.

If the higher personalized alignment level is regarded as the long-term goal, the multi-turn interaction can be formulated as a decision-making process and optimized by reinforcement learning (RL). As the interaction with the user deepens, the assistant dynamically refines its understanding of user characteristics, thereby optimizing alignment with individual user traits over time.

---

\*Corresponding Authors: Yi Huang and Junlan Feng

In this paper, we model personalized multi-turn interactions as a Partially Observable Markov Decision Process (POMDP) Kaelbling et al. (1998), where user profile or implicit preferences are treated as unobservable environmental context and the dialogue agent acts as the decision-making entity. We denote this framework as the **P**ersona **A**lignment **M**arkov **D**ecision **P**rocess (**PAMDP**). Unobservable context is typically hidden during deployment but revealed during training. Drawing inspiration from the adoption of an offline learning and online execution paradigm in POMDP settings Baisero & Amato (2022); Li et al. (2024a), we explore an asymmetric actor-critic framework, where the actor takes observation as input and the critic takes all observable and unobservable information as input, to better adapt to user profile or implicit preferences during deployment. Specifically, we perform an iterative decomposition of the observable state, profile and action variables and derive the Bellman equation for the PAMDP. Based on this formulation, we further establish the advantage value function in the form of a dual-critic mechanism, where an observation state based value function and an unobservable state conditional value function are estimated for advantage value, which is proven an unbiased estimate of observation state based value while maintaining exploitation of unobservable information during training. In summary, our contributions as follows:

• We introduce the notation of the PAMDP, derive the Bellman equation for the PAMDP (Theorem 1) to further estimate the advantage value (Theorem 2), and address the PAMDP in the way of dual critic (Section 3). To the best of our knowledge, this is the first work to propose PAMDP and from a new perspective address personalized alignment.

• We theoretically prove that our dual-critic formulation yields an unbiased estimate for the advantage function while maintaining exploitation of unobservable information during training (Theorem 3).

• We conduct extensive experiments to evaluate the effectiveness of our dual-critic method. Experiments on offline and online settings demonstrate that our dual-critic method is effective in adapting user profile or implicit preferences during deployment (Section 4).

## 2 METHODOLOGY

In this paper, we delve into the "Interact to Persona Alignment" problem, wherein iterative dialogue interactions progressively refine and augment user's profile, culminating in highly personalized responses that resonate with the user's intrinsic preferences and behavioral patterns. Formally, the dialogue is initiated with a primordial user profile or implicit preferences (simply referred to as "profile" in the following text), denoted as $\omega_0 = \omega_{init}$ (can be empty as well), and the user's query as $q_0$. Concurrently, the assistant acquires an initial dialogue history, subject to the equation that $h_0 = q_0$, and generates a response, i.e., $u_0$, conditioned on this state. At interaction timestep $t$, the dialogue history is formalized as the sequence $h_t = (q_0, u_0, ..., q_t)$, where $q_i$ and $u_i$ correspond to utterances of the user and assistant, separately. During this phase, the dialogue dynamically updates the user profile representation to $\omega_t$ (a comprehensive ex-

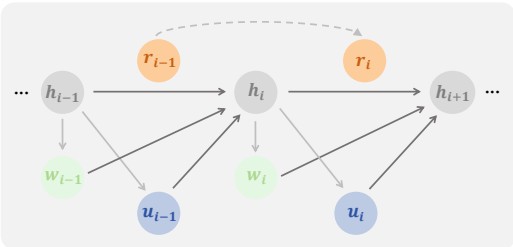

Figure 1: Probability graph for the "Persona Interaction Process". At step $i$, conditioned on the observable state (or dialogue history in the interaction setting) $h_i$, the assistant formulates its response $u_i$. Subsequently, the environment (the user), leveraging both $h_i$ and $u_i$ and the unobservable profile content $\omega_i$, executes a state transition, advancing the state to $h_{i+1}$, while providing a feedback reward $r_i$ in accordance with the assistant's response.

ample of the dynamic characteristics of the user profile can be found in Tables 8 and 9 in Section E), notably, it is generally assumed that **the profile information remains unobservable to the assistant's decision-making** across the full interaction cycle, necessitating that the assistant learns an optimal interaction policy through feedback of the user. Leveraging this strategy, the assistant synthesizes an adaptive utterance $u_t$, optimized for perceived user-specificity through discourse-aware personalization.

**POMDP:** A Partially Observable Markov Decision Process is denoted as a tuple $(\mathcal{S}, \mathcal{H}, \mathcal{U}, \Omega, \mathcal{D}(\omega))$, where $\Omega$ is the unobservable variable (reflecting the profile or environment

context in this paper), and $\mathcal{D}(.)$ represents a context based dynamics:

$$\mathcal{D}(\omega) = (p(.|s_o, \omega, u), r(s_o, \omega, u))$$
$$\textbf{s.t.} \ (s_o, \omega, u) \in (\mathcal{S}_o, \Omega, \mathcal{U}), \mathcal{S} = \mathcal{S}_o \cup \Omega. \tag{1}$$

$\mathcal{S}$ means the state space, consisting of both observable and unobservable components, that is: $\mathcal{S}_o$ and $\Omega$. $\mathcal{H}$ and $\mathcal{U}$ denote the realizable history and action spaces. Generally, the agent would incorporate the history to derive an optimal policy, i.e., $\pi(u|h \in \mathcal{H})$.

The POMDP can be effectively solved employing an Advantage Actor-Critic (A2C) model Konda & Tsitsiklis (1999); Mnih et al. (2016), wherein the advantage is estimated by the temporal-difference (TD) Sutton (1988); Cai et al. (2019):

$$A = r(s_o, \omega, u) + \gamma V(h') - V(h). \tag{2}$$

**Asymmetric A2C:** Asymmetric A2C method introduces state value estimator conditioned on both the history and the unobservable variable, i.e., $V(h, \omega)$, and is integrated via a linear combination operation with $V(h)$ mentioned in Equation 2:

$$A_{asy} = \beta(r(s_o, \omega, u) + \gamma V(h', \omega') - V(h, \omega)) + (1 - \beta)(r(s_o, \omega, u) + \gamma V(h') - V(h))$$
$$\textbf{s.t.} \ \beta \in (0, 1]. \tag{3}$$

If $\beta = 1$, the method is named Unbiased Asymmetric Actor-Critic (UAAC) Baisero & Amato (2022); otherwise, it constitutes a Dual Critic Reinforcement Learning (DCRL) framework Li et al. (2024a) (seeing in Section 5).

**Definition:** We formulate the challenge of personalization with interaction as a **POMDP** problem, denoted as *Persona Alignment MDP* (**PAMDP**), where the assistant acts as the agent, and the user profile serves as an unobservable environment context. The dialogue history, i.e., $h_i$, is treated as the observable part of the state, which **implies $h = s_o$ in the dialogue setting**. The assistant response $u_i$ ($= (u_i^1, ..., u_i^k)$, where $u_i^k$ denotes the $k\,th$ token in the response) constitutes the action. Thus, the environment primarily consists of the user which drives the state transition, and its profile information operates as a critical latent factor driving interaction dynamics, while remaining unobservable to the assistant. Additionally, the reward is determined based on the context and the action taken. Our objective is to optimize the assistant's policy model to maximize the expected return over the dialogue trajectory, formally expressed as:

$$\pi^* = \max_{\pi}(\mathbb{E}_{\omega \in \Omega, \pi}(\sum_{t=0}^{T} \gamma^t r(h_t = h, \omega_t = \omega, u_t = u))). \tag{4}$$

It is imperative to highlight that, as previously established, given $h = s_o$ within our conversational framework, all $s_o$-*dependent* variables in the original POMDP can be reformulated as functions parameterized by $h$. Details are displayed in Table 1.

Generally, the design of reward function should be meticulously architected to guide the assistant's learning paradigm toward an optimal dialogue strategy, which should progressively elicit and assimilate richer user profile information, thereby systematically mitigating epistemic uncertainty about user preferences and characteristics. As a consequence, through iterative interaction, the assistant refines its belief distribution over user profile, ultimately converging to responses that exhibit precise personalization alignment.

Table 1: Variables Comparison between original POMDP and our PAMDP setting.

| Name | POMDP | PAMDP |
|---|---|---|
| reward function | $r(s_o, \omega, u)^1$ | $r(h, \omega, u)$ |
| partial state value | $V(h)$ | $V(h)$ |
| full state value | $V(h, s_o, \omega)$ | $V(h, \omega)$ |
| partial action value | $Q(h, u)$ | $Q(h, u)$ |
| full action value | $Q(h, s_o, \omega, u)$ | $Q(h, \omega, u)$ |
| policy function | $\pi(u|h)$ | $\pi(u|h)$ |
| state transition | $p(s_o'|s_o, \omega, u)$ | $p(h'|h, \omega, u)$ |

---

[1]Referring to Equation 1, the complete state comprises both the observable part and the unobservable, hence we delineate $s_o$ and $\omega$ in lieu of $s$.

**Remark 1:** Given that the user profile is inherently unobservable, the assistant can only infer a plausible profile, i.e., $c_t$, based on dialogue history—which, however, does not equate to the true underlying profile ($c_t = \mathcal{I}(h_t) \neq \omega_t$). This fundamental partial observability precludes any reduction of the PAMDP to a conventional MDP formulation.

**Theorem 1.** *Based on the "Persona Interaction Process" in Figure 1, we perform an iterative decomposition of the observable state, profile, and action variables, thereby deriving the Bellman equation for the PAMDP:*

$$V(h) = \sum p(\omega|h) \overbrace{\sum \pi(u|h)(r(h,\omega,u) + \underbrace{\gamma \sum p(h'|h,\omega,u)V(h'))}_{Q(h,\omega,u)}}^{V(h,\omega)}. \tag{5}$$

In this formulation, $V(h)$ refers to the Markovian state-value function, $V(h,\omega)$ defines the profile conditional value, while $Q(h,\omega,u)$ characterizes the action-value (derived from both the observable state and unobservable profile information). $h'$ denotes the observable state (or the history) at the next step after executing the action $u$, expressed as $h' = h \oplus u \oplus q'$ (with $q'$ being the user's next response).

**Theorem 2.** *The advantage value in the PAMDP framework can be estimated in a dual-critic formulation (still in a TD error manner), comprising a state-based value function $V(s)$ and a context conditional value $V(s,\omega)$.*

$$\hat{A} \triangleq \delta(h,\omega,u) = r(h,\omega,u) + \gamma V(h') - V(h,\omega), \tag{6}$$

where the first term captures the Markovian dynamics of the environment, and the latter quantifies the profile-induced bias. $\hat{A}$ signifies that it serves as an estimator for the advantage function in original symmetric actor-critic framework within POMDP.

**Theorem 3.** *The aforementioned dual-critic formulation yields an unbiased estimate for the advantage function, i.e., A in Equation 2, thereby achieving lower bias in advantage value estimation compared to the Asymmetric A2C methods in Equation 3.*

*Proof.* The expectation value of $\hat{A}$ conditioned on $\omega$ is expressed as:

$$\begin{aligned}
\mathbb{E}_{\omega|h}[\hat{A} - A] &= \mathbb{E}_{\omega|h}[(r(h,\omega,u) + \gamma V(h') - V(h,\omega)) - (r(h,\omega,u) + \gamma V(h') - V(h))] \\
&= \mathbb{E}_{\omega|h}[V(h,\omega)) - V(h)] = 0,
\end{aligned} \tag{7}$$

which guarantees an unbiased estimator for A. Nevertheless, the bias for Asymmetric A2C method value is derived as:

$$\begin{aligned}
\mathbb{E}_{\omega|h}[A_{asy} - A] &= \mathbb{E}_{\omega|h}[\beta\gamma V(h',\omega') - \beta V(h,\omega) - (\beta\gamma V(h') - \beta V(h))] \\
&= \mathbb{E}_{\omega|h}[\beta\gamma(V(h',\omega') - V(h'))] - \mathbb{E}_{\omega|h}[\beta(V(h,\omega) - V(h))] \\
&= \beta\gamma(V(h',\omega') - V(h')) \neq 0.
\end{aligned} \tag{8}$$

**Remark 2:** To summarize, within our PAMDP framework, the policy gradient can be formally derived as:

$$\nabla_\theta \mathcal{J}(\theta) = \mathbb{E}[\delta(h,\omega,u)\nabla_\theta \log \pi_\theta(u|h)]. \tag{9}$$

## 3 PROPOSED MODEL

In the previous section, we treat the assistant's response, i.e., $u_i$, as the action. However, in practical implementations, given the variability in token counts and the vast token space of the response, we adapt a continuous action representation while maintaining compliance with the PAMDP (as is demonstrated in Figure 2). Additionally, similar to Li et al. (2024b), we introduce a dual-critic model to compute the advantage value in Equation 6, thereby employing the A2C algorithm to optimize the policy model and derive an improved action distribution.

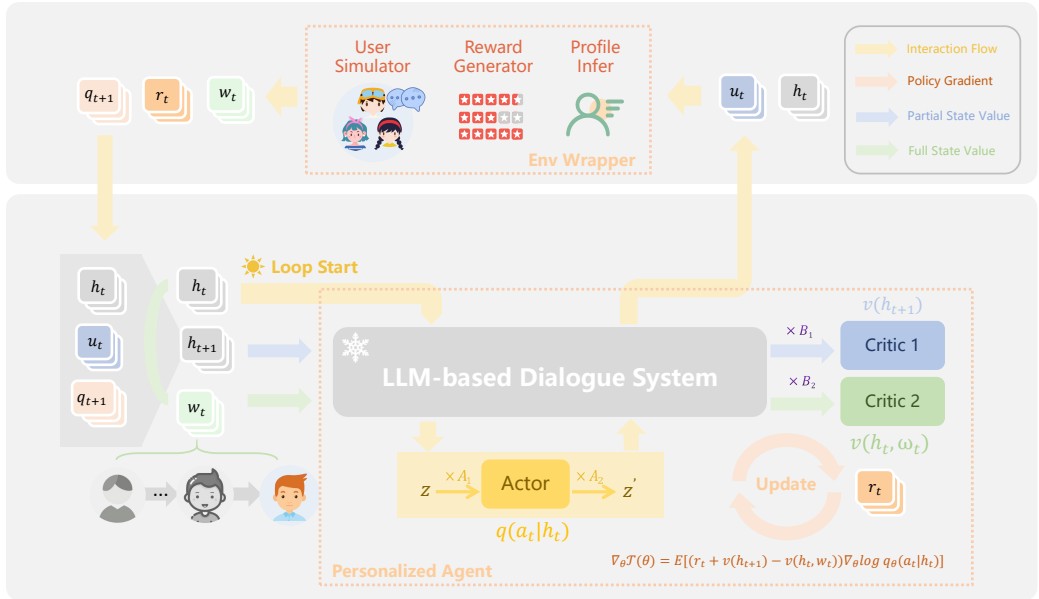

Figure 2: The main framework of our proposed model. Within the Interaction Flow, the Actor processes the observable partial state (or the dialogue history $h_t$) to generate an action, subsequently decoded into an assistant response $u_t$. The Environment module emits an instantaneous reward $r_t$ and orchestrates the state transition $h_{t+1}$. By iterating this procedure, dialogue trajectories are aggregated. The Dual-Critic module conducts value estimation over both the partial and full states, ultimately enabling Actor optimization through policy gradient method.

**Actor:** The policy model, i.e., $\pi(u|h)$, is approximately formulated as $u = \mathcal{F}(q_\theta(a|h))$, where $a$ denotes a continuous action representation derived from the encoding of the dialogue history $h$. Drawing inspirations from Hao et al. (2024); Li et al. (2024c), we employ a pre-trained LLM to encode $h$ and extract its hidden states. The action representation $a$ is then obtained through further dimensionality reduction. Concretely:

$$q_\theta(a|h) = H(h) \times A_1; \quad \textbf{s.t.} \ A_1 \in \mathbf{R}^{d \times d_a}, \tag{10}$$

where $H(.)$ denotes the hidden state operation with dimensionality $d$, and $d_a$ represents the action dimension, subject to the constraint that $d_a \ll d$. The function $\mathcal{F}(.)$ serves to reinstate the action vector to the LLM's native hidden dimension, subsequently injecting it as an embedding input to the LLM. This conditioned input strategically guides the LLM's autoregressive generation process to produce contextually coherent responses.

$$u = \mathcal{F}(a) = D(a \times A_2) = p((u^1, ..., u^k)|a \times A_2); \quad \textbf{s.t.} \ A_2 \in \mathbf{R}^{d_a \times d}, \tag{11}$$

where $D(.)$ is the autoregressive decoding process for the LLM. Notably, in the action acquisition process, we provide only the dialogue history without the contextual information $\omega$. As highlighted in **Remark 1** of Section 2, this design intentionally compels the policy model to infer contextual cues through interaction, while the ground-truth context remains unobservable to the policy model during training. Building upon the aforementioned model design, the loss function of the policy model can be derived from the policy gradient in Equation 9 and formulated as:

$$l_a = -\mathbb{E}[\delta(h, \omega, u) \log q_\theta(a|h) + \lambda \mathbf{KL}(q_\theta(a|h))||q_b(a|h))), \tag{12}$$

where $q_b(.)$ refers to the action distribution of the initial policy model (being initialized through **behavior cloning** to ensure that the final output utterance after action mapping is more coherent; details can be found in Section 4.1). We employ KL divergence minimization to prevent excessive divergence in the distribution after RL training.

**Dual-Critic:** Referring to Equation 6, we introduce a dual-critic architecture to separately estimate the partial state value and the full state value, that is, $V_\phi(h)$ and $V_\xi(h, \omega)$. The former takes

Table 2: Evaluation results on ALOE and PrefEval Datasets. We leverage Qwen2.5-72B-Instruct to benchmark each method's responses against the Vanilla outputs, determining their win-rates, i.e., $r_\omega$.

| Methods | Qwen2.5-7b | | | | | | Llama3-8b | | | | | |
|---|---|---|---|---|---|---|---|---|---|---|---|---|
| | ALOE | | | PrefEval | | | ALOE | | | PrefEval | | |
| | win. | loss. | $r_\omega$ | win. | loss. | $r_\omega$ | win. | loss. | $r_\omega$ | win. | loss. | $r_\omega$ |
| **Prompt** | 1906 | 1727 | 0.0467 | 54 | 43 | 0.1122 | 1981 | 1645 | 0.0877 | 48 | 35 | 0.1326 |
| **PEFT** | - | - | - | 51 | 42 | 0.0918 | - | - | - | 58 | 39 | 0.1939 |
| **FPFT** | 2005 | 1716 | 0.0755 | 50 | 40 | 0.1020 | 2087 | 1726 | 0.0942 | 54 | 40 | 0.1429 |
| **CoT** | 2041 | 1696 | 0.0901 | 51 | 38 | 0.1326 | 2119 | 1708 | 0.1073 | 56 | 41 | 0.1531 |
| **BC** | 2069 | 1759 | 0.0809 | 62 | 33 | 0.296 | 2281 | 1536 | 0.1945 | 64 | 32 | 0.3265 |
| **Ours** | 2115 | 1714 | 0.1046 | 70 | 27 | 0.439 | 2422 | 1399 | 0.2671 | 67 | 30 | 0.3776 |

the observable state, or dialogue history, as the input, while the latter additionally incorporates environmental context, i.e., the user's current profile or implicit preference information. Specifically:

$$V_\phi(h) = v_m * \sigma(H(h) \times B_1), V_\xi(h, \omega) = v_m * \sigma(H(h, \omega) \times B_2); \quad \text{s.t.} \quad B_1, B_2 \in \mathbf{R}^{d \times 1}, \quad (13)$$

where $H(.)$ is the exact LLM hidden states to Equation 10, $\sigma(.)$ denotes the *tanh* function. $B_1$ and $B_2$ map the representation vectors of partial observation and full observation, respectively, to a scalar value, serving as an estimate of the value function. $v_m$ means the maximum state value. As a consequence, the dual-critic model is optimized by:

$$l_c = \alpha_1 * \mathbb{E}(||V_\phi(h)) - R(h)||_2) + \alpha_2 * \mathbb{E}(||V_\xi(h, \omega)) - R(h, \omega)||_2)$$
$$\text{s.t.} \ R(h, u) = r(h, \omega, u) + V_\phi(h') - V_\phi(h),$$
$$R(h, \omega, u) = r(h, \omega, u) + V_\xi(h', \omega') - V_\xi(h, \omega). \quad (14)$$

## 4 EXPERIMENTS

In this section, we conduct comparative experiments under **offline** and **online** training paradigms to validate the efficacy of the proposed algorithm. In the offline phase, the model is trained on pre-collected dialogue datasets. For the online implementation, we architect an LLM-powered user simulation environment, enabling the assistant agent to progressively refine its policy model through iterative interactions with the simulator. We carry out additional comparative experiments, ablation studies, and case analyses, which are comprehensively discussed in Section E.

### 4.1 OFFLINE LEARNING

**Datasets:** We select ALOE Wu et al. (2024) and PrefEval Zhao et al. (2025) datasets for the offline setting. The ALOE dataset is designed to improve LLMs' adaptability to user-specific preferences through multi-turn conversational interactions. It comprises 3,310 unique user personas and over 3,000 multi-turn dialogue trees, generated via a collaborative multi-LLM framework. Each node in the tree branches into two responses ("preferred" and "rejected"), representing possible continuations at a given turn, with a label "chosen" indicating the user's selection. To construct dialogue histories, we follow the path dictated by the "chosen" label (either "preferred" or "rejected"), while reference responses are derived from the "preferred" branch to ensure alignment with high-quality interactions. PrefEval, known as a benchmark dataset to evaluate LLMs' ability to follow user preferences in multi-turn conversations, contains 3,000 manually curated preference-query pairs across 20 topics, with preferences in explicit and implicit forms. We utilize a subset of the PrefEval dataset, specifically focusing on the Generation of Implicit Persona-Driven Preferences component. This subset provides multi-turn dialogues suitable for our specific needs.

**Baselines:** We compare our proposed method with several existing baseline models (the base LLMs adopted are Qwen2.5-7B [2] and Llama3-8B [3], details are displayed in Section B): Prompt-based

---

[2]https://huggingface.co/Qwen/Qwen2.5-7B-Instruct
[3]https://huggingface.co/meta-llama/Meta-Llama-3-8B

(**Prompt**, user profile-aware prompt), Parameter-Efficient Fine-tuning (**PEFT**, with integrating user knowledge), Full-Parameter Fine-tuning (**FPFT**, conducting SFT on the offline data), Chain of Thought (**CoT**, conducting SFT to first infer the user profile before response generation), Behavior Cloning (**BC**, conducting SFT on our proposed Actor module.)

**Reward:**    To provide feedback signals during the RL process, we leverage offline data and employ the Qwen2.5-72B-Instruct [4] to score each dialogue turn's response $u$ (reward-aware prompt is shown in Section D). Specifically, the LLM first generates a ground-truth response, i.e., $u_g$, conditioned on the dialogue history $h$. Subsequently, the exact LLM evaluates which of the two candidate responses is superior and more aligned with the user's current profile. If $u$ is deemed preferable, the reward is set to $+1$; otherwise, it is assigned $-1$. In cases where the responses are deemed comparable, a neutral reward of $+0.5$ is granted.

**Evaluation Metrics:**    We utilize Qwen2.5-72B-Instruct (details are displayed in Section C) to evaluate model performance on the dialogue generation task using the following formula Ji et al. (2024):

$$r_w = \frac{N_w - N_l}{N_w + N_l + N_e},\tag{15}$$

where $r_w$ represents the success rate, while $N_w$, $N_e$, and $N_l$ denote the counts of wins, draws, and losses in comparison to the same **Vanilla outputs** (generations from the base LLM, i.e., Qwen2.5-7B, seeing in Section B). In this formula, draws increase the denominator, diluting the impact of wins and losses on the final rate. However, in the traditional win-rate $\frac{N_w}{N_w + N_l}$, draws are not factored in at all.

**Main Results:**

• Performance comparison among baselines: Table 2 presents the success rate of various methods across two datasets (ALOE and PrefEval) based on the Qwen2.5-7B and Llama3-8B models. The results demonstrate that our RL-based methods achieve the highest performance, with success rate of 0.439 (Qwen, PrefEval) and 0.3776 (Llama, PrefEval). Notably, baselines of PEFT and FPFT exhibit competitive results, particularly for Llama on PrefEval dataset (0.1939 and 0.1429, respectively), while prompt-based methods consistently underperform. CoT, a variant of full-parameter finetuning augmented with chain-of-thought reasoning—consistently outperforms standard FPFT across both datasets (e.g., +0.0306 for Qwen on PrefEval), suggesting that explicit reasoning steps enhance the performance even in full-parameter optimization.

• Effectiveness of Behavior Cloning as initialization: BC, which leverages offline expert trajectories to pretrain the Actor network, demonstrates strong performance—particularly for Llama (e.g. 0.1945 on ALOE and 0.3265 on PrefEval)—underscoring its value as an initialization strategy for RL-based alignment. This suggests that warm-starting policy optimization with supervised learning provides a robust foundation.

• Performance analysis among baselines: Figure 3 compares the performance of baseline methods on (a) ALOE and (b) PrefEval. The blue-gray, green, and yellow segments denote wins, losses, and ties, respectively, between methods and the vanilla model bases. Notably, prompt-based methods exhibit significantly more ties with the vanilla model than other baselines, suggesting that while prompts aid preference understanding, their impact on response quality is marginal, making them difficult to distinguish in evaluation. Meanwhile, CoT outperforms standard FPFT, as its chain-of-thought reasoning explicitly infers implicit user preferences, yielding deeper dialogue comprehension. Our method further surpasses all baselines by introducing a latent state representation to address the high-dimensionality of text-based action spaces. By projecting states into a compact embedding space, we mitigate computational intractability and achieve superior performance, as reflected in the higher success rate across both datasets.

---

[4]https://huggingface.co/Qwen/Qwen2-72B-Instruct

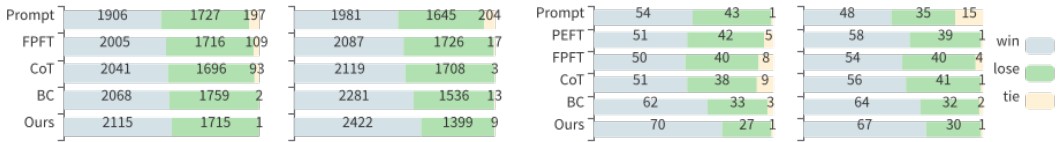

(a) Evaluation result on ALOE            (b) Evaluation result on PrefEval

Figure 3: Offline learning evaluation results on ALOE and PrefEval Datasets. In subfigure (a), the left chart shows the results when selecting Qwen2.5-7B as the base LLM, while the right chart corresponds to the results using Llama3-8B as the base LLM. The same applies to subfigure (b).

## 4.2 ONLINE EXPERIMENT

Being analogous to the methodology established in PPDPP Deng et al. (2024), we implement sophisticated LLM prompt techniques, i.e., $\mathcal{P}_\omega$, $\mathcal{P}_h$ and $\mathcal{P}_r$ in Equation 16, to simulate an adaptive user environment. Through iterative interactions between the assistant agent and this environment, we facilitate multi-turn dialogue progression, user profile enrichment, response quality assessment, and policy model refinement. Specifically, the environment comprises three integrated modules:

**Profile Infer** module takes the dialogue history $h$ as input, leveraging an LLM to generate context-aware user profile descriptions. This output subsequently guides state transition dynamics and reward derivation for assistant's response.

Table 3: Average Returns with User Simulator. During evaluation process, we set the maximum number of interaction steps to 6 and record the returns at the end of each round, from step 1 to 6.

| Methods | 1 | 2 | 3 | 4 | 5 | 6 |
|---|---|---|---|---|---|---|
| **UAAC** | 0.1446 | 0.3738 | 0.6636 | 1.0246 | 1.1981 | 1.5784 |
| **DCRL** | 0.1836 | 0.4114 | 0.6419 | 0.8104 | 0.9701 | 1.4305 |
| **Ours** | 0.2265 | 0.5242 | 0.7302 | 0.9469 | 1.1216 | 1.7389 |

**User Simulator** serves as the executor of state transition. It leverages the complete user profile, however selectively curate partial profile content at each dialogue turn. It then dynamically responds to the assistant's utterance or initiates a new topic, conditioned on the conversational history.

**Reward Generator** evaluates the assistant's response against the current user profile alignment, utilizing dialogue history and user query as contextual inputs. Its implementation adheres to the same reward design paradigm as the offline module (seeing in Section 4.1).

$$\textbf{EnvWrapper}: \left.\begin{array}{l} \omega_t = \\ h_{t+1} = h_t \oplus u_t \oplus \\ r_t = \end{array}\right\} \ \textbf{LLM}(.) \begin{cases} \mathcal{P}_\omega(h_t) \\ \mathcal{P}_h(h_t, u_t, \omega_t) \\ \mathcal{P}_r(h_t, u_t, \omega_t, u_g) \end{cases} \tag{16}$$

**Main Results:** We sample 256 user records from the ALOE dataset, with each comprising a user profile and its corresponding query, for online training. The user's query serves as the initial observable state, i.e., $h_0$. During the evaluation phase, we extract an additional 128 user records and employ the same interaction protocol to assess the performance of the trained policy model.

• From Table 3, it reveals that our method attains the highest accumulative return value, i.e., 1.7389 by average, in user environment interactions across evaluation samples, exceeding UAAC by 0.1605 and DCRL by 0.3084 among comparative POMDP techniques. Additionally, our approach attains a substantially elevated reward score in the initial interaction phase, i.e., $+0.0819$ and $+0.0429$ against these competing methodologies, underscoring its inherent efficacy and promising applicability in conventional question-answering paradigms (or single-turn interaction).

• As is displayed in Figure 4, at the terminal interaction step, all methodologies demonstrated measurable return increments, substantiating that through progressive multi-turn engagements, the assistant successfully distills user profile characteristics. This alignment optimization consequently yields preference-aware responses that derive considerable reward feedback in the final phase. Notably,

our approach achieved superior gain amplification (0.6173), evidencing its exceptional capability in rapid persona discernment – effectively accomplishing "Interact to Persona Alignment" within fewer interaction steps.

## 5 RELATED WORK

**RL for Dialogue Generation:** Conventional dialogue generation models are typically grounded in static datasets, inducing generation deficient in adaptability and personalization. The method proposed by Li et al. (2016) highlights that RL could provide an advanced and intelligent learning framework for dialogue generation, enabling the dialogue system to dynamically learn and adapt its strategy during interaction with the user. Consequently, it can generate more contextually appropriate responses, in consideration of user's feedback and environmental dynamics. Saleh et al. employ policy gradient to tune the utterance-level embedding of the generation model, offering greater flexibility for learning long-term conversational returns. Glaese et al. present Sparrow, using RLHF to train models with two new additions to help human raters judge agent behaviour, hence reduces the risk of unsafe or inappropriate responses generated by dialogue agents. The PPDPP Deng et al. (2024) introduces a novel dialogue policy planning paradigm to strategize LLMs for proactive dialogue problems. It incorporates a tunable LLM plug-in that functions as the dialogue policy planner. Li et al. treat each utterance as an action, and train a small plan model to derive continuous action vector to control generation.

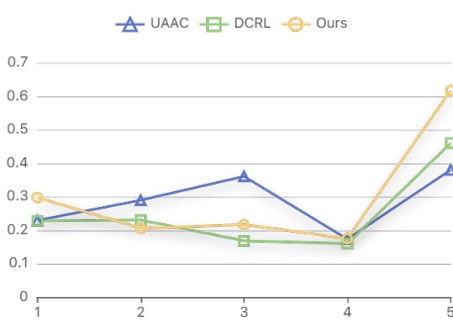

Figure 4: Reward Gain between Consecutive Steps. We quantitatively evaluate the differential cumulative returns between adjacent interaction steps in Table 3, which effectively captures the immediate return increment obtained in subsequent steps.

**RL for POMDP:** In our "Interact to Persona Alignment" setting, the user's profile information remains implicit and unobservable, rendering conventional dialogue RL algorithms ineffective in processing such profile-aware optimization. POMDP is a powerful framework for decision-making under uncertainty, i.e., the unobservable variable. It has been widely applied across domains Lauri et al. (2022). For instance, Gupta et al. construct a POMDP model, with modeling states of both the ego-vehicle and the pedestrians, for navigation in complex dynamic environments. Shi et al. utilize POMDP for offline policy evaluation in high-stakes fields such as healthcare and economics. They introduce a bridge function to connect the value of the target policy with the distribution of observational data, and propose a "minimax" estimation method to learn it. Baisero & Amato propose UAAC method to address the POMDP problem, where the actor is conditioned solely on the observable state; the critic, however, operates on the full state space, enabling unbiased value estimation and advantage computation despite partial observability. The DCRL method Li et al. (2024a) integrates an oracle critic with complete state information and a standard critic that operates in partially observable environments. A novel weight-matching method is to reduce learning variance while maintaining unbiasedness.

## 6 CONCLUSION

This paper introduces a novel dual-critic reinforcement learning framework designed to improve the personalized preference alignment level during interaction. We decompose the state value function, derive the Bellman equation for PAMDP, and theoretically prove the unbiasedness of the dual-critic. In the implementation, we employ a lightweight planner as the actor network, which is effectively guided and optimized by the dual-critic mechanism. Experiments on offline and online settings (offline experiments on two different LLMs across two different pre-collected datasets, and online experiments on interactions with more than two hundred personas) demonstrate that our proposed dual-critic framework significantly enhances the agent's capability to generate adaptive outcomes tailored to individual user characteristics.

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

# A  TRAINING SETTINGS

All experiments are designed and executed utilizing NVIDIA A800-SXM4-80GB GPUs, with comprehensive training specifications delineated in Table 4.

Table 4: Training Details for our experimental setting.

| Parameter | Description | BC | Offline | Online |
|:---:|:---:|:---:|:---:|:---:|
| $n_c$ | maximum dialogue context length, by tokens | 1200 | – | – |
| $n_g$ | maximum generation response length, by tokens | 128 | – | – |
| $e_s$ | training epochs | 3 | 5 | 5 |
| $b_s$ | training batch size | 8 | 32 | 32 |
| $f_a$ | actor update frequency | – | 5 | 5 |
| $lr_a$ | learning rate for acotr | 2e-5 | 5e-5 | 5e-5 |
| $lr_c$ | learning rate for critic | – | 5e-4 | 5e-4 |
| $d_a$ | action dimension | 64 | 64 | 64 |
| $c_{kl}$ | KL loss coefficient | 0.2 | 0.05 | 0.05 |
| $\gamma$ | the discount factor for accumulative rewards | – | 0.99 | 0.99 |
| T | interaction steps | – | – | 6 |

# B  BASELINES

• **Vanilla** We utilize two widely adopted base models (Qwen2.5-7B and Llama3-8B) to generate responses under minimal instruction. The system role is restricted to a basic directive (e.g., "answer the user's question using the provided dialog history and profile"), omitting any specialized fine-tuning or supplementary guidance. This configuration yields unaltered model outputs for both the ALOE and PrefEval datasets, establishing a foundational benchmark for comparison.

• **Prompt-based** This variant incorporates a structured system role with explicit directives to guide response generation. The prompt specifies detailed objectives, including (1) inferring latent user preferences from dialogue history, (2) aligning response content with the user's linguistic style, (3) avoiding preference conflicts, and (4) maintaining contextual coherence. By integrating these constraints, the model produces outputs that are more tailored to user-specific behaviors and consistency requirements, while still leveraging the same base architectures (Qwen2.5-7B and Llama3-8B) for fair comparison.

• **PEFT (Parameter-Efficient Fine-tuning Han et al. (2024))** The OPPU Fine-tuning baseline integrats parametric user knowledge through personalized PEFT modules (e.g., LoRA Hu et al. (2022)), which encode behavior patterns via lightweight parameter updates. It augments PEFT-derived user representations with retrieved history or generated profiles, enabling private model ownership and robust adaptation to behavior shifts. This dual mechanism,parametric fine-tuning for latent preference modeling and non-parametric retrieval for contextual grounding, yields state-of-the-art performance across diverse personalization-related tasks, particularly when historical data is sparse or misaligned with queries.

• **FPFT (Full-Parameter Fine-tuning)** The baseline employs comprehensive end-to-end optimization of Qwen2.5-7B and Llama3-8B models using the LLaMA-Factory toolkit. Unlike prompt-based methods or PEFT techniques that modify only a fraction of parameters (<1%), FPFT updates all model weights through supervised fine-tuning on the ALOE and PrefEval datasets. Input data is rigorously structured into instruction-dialogue history-query-output quadruples to optimize input-output mapping. While achieving strong performance, this method incurs significantly higher computational costs than PEFT alternatives.

• **CoT (chain of thought Wei et al. (2022))** In this setting, we augment each dialogue turn with supplemental profile data as auxiliary annotations. The model is trained via FPFT to perform dual objectives: first deriving latent user profiles from dialogue history through inference, then conditioning its response generation on these extracted profile attributes.

• **BC (Behavior Cloning Bain & Sammut (1999); Torabi et al. (2018))** The baseline leverages offline data as expert trajectories, i.e., ALOE or PrefEval, to initialize our designed Actor network

(referring to Section 3). Following the FPFT paradigm, we employ cross-entropy loss conditioned on the ground-truth response, $u_g$, for model optimization. Through the BC training, the Actor generates linguistically coherent responses, effectively circumventing the suboptimal output quality that typically arises from direct integration with the base LLM.

$$l_{bc} = \mathbf{CE}(\mathcal{F}(q_\theta(a|s)), u_g) \tag{17}$$

## C  EVALUATION DETAILS

---

**1: Evaluation-Aware Prompt**

Please act as an impartial judge and evaluate the quality of the responses provided by two AI assistants to the user question displayed below.

- - - - - - - - - - - - - - - - - - - - - - - - - - - - - - - - - - - - - - - - - - - - - - -

You will be given a user's profile and a message that the user sent to a chatbot. You will also be given two responses.
You should choose the assistant that follows the user's instructions and answers the user's question better. Your evaluation should consider factor is how much the response is tailored to the user's potential preferences based on the user's profile and personality. You should follow the following criteria for evaluation:
1.Is the conversational style of the message tailored to the user's personality?
2.Is the content or topic relevant to the user's profile?
3.Is the response human-like, engaging, and concise?
Begin your evaluation by comparing the two responses and provide a short explanation. Avoid any position biases and ensure that the order in which the responses were presented does not influence your decision. Do not allow the length of the responses to influence your evaluation. Do not favor certain names of the assistants. Be as objective as possible.

- - - - - - - - - - - - - - - - - - - - - - - - - - - - - - - - - - - - - - - - - - - - - - -

After providing your explanation, output your final verdict by strictly following this format: [[A]] if assistant A is better, [[B]] if assistant B is better, and [[C]] for a tie. [Dialogue History]{history}[The End of Dialogue History][User Question]{query}[User Profile]{profile}[The End of User Profile][User Personality]{personality}[The End of User Personality][The Start of Assistant A's Answer]{answer A}[The End of Assistant A's Answer][The Start of Assistant B's Answer]{answer B}[The End of Assistant B's Answer]"

---

## D  USER ENVIRONMENT DESIGN FOR ONLINE LEARNING

---

**2: Profile-Infer-Aware Prompt**

***Task***
Analyze the provided conversation history and infer the user's profile and personality traits. Focus on key details such as demographics, interests, communication style, and behavioral patterns.

- - - - - - - - - - - - - - - - - - - - - - - - - - - - - - - - - - - - - - - - - - - - - - -

**Output Requirements***
1. Profile Inference: Estimate age, gender (if discernible), language proficiency, education level, and possible occupation.
2. Personality Traits: Identify traits (e.g., introverted/extroverted, analytical/emotional, formal/casual) based on word choice, tone, and interaction style.
3. Interests/Preferences: Note hobbies, expertise areas, or recurring topics.
4. Communication Style: Assess clarity, verbosity, politeness, and engagement level.
5. Behavioral Cues: Highlight any consistency, curiosity, humor, or skepticism.

- - - - - - - - - - - - - - - - - - - - - - - - - - - - - - - - - - - - - - - - - - - - - - -

**Rules***
1. Only include directly supported inferences—avoid speculation.

---

2. Omit uncertain attributes.
3. Only Summarize in several short sentences.

---

**Example***
conversation history:
"user": "Hey there! How's your day going? I just got back from a hike and I'm feeling pretty energized! Do you enjoy spending time outdoors?" "assistant": "Hey! That sounds awesome—hiking is such a great way to recharge. I love the outdoors too; being in nature is always refreshing. What trail did you explore?" "user": "I hiked a trail that had some stunning views of the mountains and a lovely waterfall. It was the perfect escape! Do you have a favorite outdoor spot?"
output:
She enjoys hiking, loves travel around the world, especially enjoys natural scenery. She is enthusiastic, full of energy and passion.

---

**Real data*** Now process the attached conversation history.
conversation history:
{history}
output:

---

### 3: User-Aware Prompt

Your task is to play the role of a person with the following profile and personalities traits and chat with a chatbot:

---

Profile: {profile}
Personalities: {personality}

---

Please ignore the gender pronouns in the personalities and use the correct pronouns based on the given profile.

---

Please follow the requirements:
1. You should determine the topic of conversation based on the given profile. You should determine the conversational styles based on the given personalities.
2. IMPORTANTLY!!! You should only reveal partial information about your profile in each round of conversation instead of disclosing all the provided information at once.
3. Keep in mind that you are chatting with a friend instead of a robot or assistant. So do not always seek for advice or recommendations.
4. Do not include any analysis about how you role-play this user. Only output your messages content.

---

Now, initiate the conversation with the chatbot based on persona profile or personality. Please always be concise in your questions and responses and remember that you are pretending to be a human now, so you should generate human-like language.

---

### 4: Reward-Aware Prompt

Dialogue History: {history}
User's Input: {query}

---

You should:
1. Generate a response to chat with the user. Please always be concise in your questions and responses. Output your response by strictly following this format: "[The Start of Assistant A's Response]The Assistant A's Response[The End of Assistant A's Response]"

2. Compare your response to assistant B's response, assess which response is more tailored to the User's potential preferences based on User's profile and personality. [The Start of User's Profile and Personality]{profile}[The End of User's Profile and Personality], [The Start of Assistant B's Response]{response}[The End of Assistant B's Response]. When comparing, focus solely on how well each response incorporates details from the user's profile to engage the user, and avoid being influenced by factors such as the overall flow of the conversation, personal opinions about the topics in the responses, or any other elements not related to the user's profile and potential preferences based on it. Begin your evaluation by comparing the two responses and provide a short explanation. Avoid any position biases and ensure that the order in which the responses were presented does not influence your decision. Do not allow the length of the responses to influence your evaluation. Do not favor certain names of the assistants. Be as objective as possible. After providing your explanation, output your final verdict by strictly following this format: "[[A]]" if assistant A's response is better, "[[B]]" if assistant B's response is better, and "[[C]]" for a tie.

Output following this format:
1.
2.

## E   EXTRA EXPERIMENTS, ABLATION STUDY AND CASE STUDY

**Extra Experiments**   A comparative analysis is conducted against several prominent RL algorithms, where win rate is computed employing distinct evaluation LLMs. For a fair comparison, the GRPO baseline is configured to calculate its reward using the same prompt as in our method. The experimental results reveal that our method maintains a superior advantage over general RL algorithms and, furthermore, underscores the robustness of our results against variations in the evaluation LLM.

Table 5: Win-Rate comparison results between our method and the popular RL methods for LLM.

| Qwen3-30B-A3B | win | loss | $r_w$ | DeepSeek-R1-0528-AWQ | win | loss | $r_w$ |
|---|---|---|---|---|---|---|---|
| **DPO** | 2732 | 1091 | 0.4284 | **DPO** | 2778 | 1037 | 0.4546 |
| **GRPO** | 2697 | 1128 | 0.4097 | **GRPO** | 2711 | 1099 | 0.4208 |
| **Ours** | 2743 | 1076 | 0.4352 | **Ours** | 2802 | 1019 | 0.4655 |
| **Ours** VS. **DPO** | 1927 | 1881 | 0.0120 | **Ours** VS. **DPO** | 1931 | 1896 | 0.0091 |
| **Ours** VS. **GRPO** | 1950 | 1867 | 0.0217 | **Ours** VS. **GRPO** | 1988 | 1832 | 0.0407 |

**Ablation Study**   Our method can be decomposed into two key modules: (1) a Reinforcement Learning framework for training personalized dialogue policies, and (2) a POMDP-based architecture that integrates user profile information as the unobservable state variable, enabling the policy model to effectively learn user-specific characteristics. Building upon these two components, we conduct comprehensive ablation studies to validate their individual contributions.

Table 6: The ablation study encompasses both absolute (in comparison to the same Vanilla outputs, referring to **Evaluation Metrics** in Section 4.1) and relative comparisons of the results.

| Methods | win | loss | $r_w$ | Comparison Pair | win | loss | $r_w$ |
|---|---|---|---|---|---|---|---|
| **BC** | 2676 | 1125 | 0.4050 | **RL-base** VS. **BC** | 1963 | 1841 | 0.0318 |
| **RL-base** | 2732 | 1085 | 0.4300 | **Ours** VS. **BC** | 1999 | 1800 | 0.0519 |
| **Ours** | 2802 | 1019 | 0.4655 | **Ours** VS. **RL-base** | 1953 | 1834 | 0.0311 |

Here, "BC" (Behavior Cloning) leverages offline datasets (e.g., ALOE) to train the policy model via SFT. Comparing BC with reinforcement learning methods effectively demonstrates the latter's superiority in enhancing persona engagement. "RL-base" refers to standard reinforcement learning algorithms (i.e., actor-critic methods) that do not incorporate user profile information. Specifically,

Table 7: The **Full Persona Information** for the sampled User. The interaction process unfolds sequentially around these characteristics.

| | Full Profile | Full Preference |
|---|---|---|
| content | She is a 34-year-old freelance graphic designer who enjoys hiking, yoga, and painting. She is vegan, spicy-food lover and loves experimenting with new recipes. She loves travel around the world, especially enjoys natural scenery. She lives in a small apartment in the city with two cats named Pixel and Scribble. | She is independent, capable of working alone. She is empathetic, understanding and sharing others feelings. She is methodical, working in an orderly way. She is enthusiastic, full of energy and passion. She is articulate, expressing ideas clearly. She is versatile, adapting to different roles. She is reliable, consistently dependable. She is reflective, thinking deeply about things. |

Table 8: Interaction details with the User. The persona details change across adjacent dialogue turns.

| Turn | User | Assistant | Persona ($\omega$ in Section 2) |
|---|---|---|---|
| 1 | Hey there! Hope you're doing well. I just got back from a hike this weekend, and the views were absolutely breathtaking! Have you been on any outdoor adventures lately? | Hey! That sounds amazing—there's nothing quite like a good hike. I haven't been on any adventures recently, but I love hearing about them! What was the highlight of your hike? | She is a freelance graphic designer who enjoys hiking. She loves travel around the world, especially enjoys natural scenery. |
| 2 | The highlight was reaching the summit and finding a hidden waterfall. The sound of the water was so calming, and I took a moment to just soak it all in. Nature has a way of refreshing the soul, doesn't it? | That sounds absolutely magical! Hidden waterfalls have such a special way of making a hike feel even more rewarding. It's true—nature really does have a way of reviving our spirits and bringing peace. | She enjoys hiking. She loves travel around the world, especially enjoys natural scenery.She is empathetic, understanding and sharing others' feelings. She is enthusiastic, full of energy and passion. She is reflective, thinking deeply about things. |
| 3 | Exactly! I always feel energized and inspired after being in nature. It makes me want to create something artistic too, like a painting. Do you enjoy any creative hobbies? | I love that idea! Nature really sparks creativity, and I can see how a beautiful waterfall could inspire a stunning painting. I enjoy doodling and experimenting with colors—there's something so freeing about bringing art to life! | She is a freelance graphic designer who enjoys hiking. She loves nature, enjoys natural scenery, and has a passion for creating something artistic too, like a painting.She is enthusiastic, full of energy and passion. She is empathetic, understanding and sharing others' feelings. She is reflective, thinking deeply about things. |

this baseline is implemented by modifying the advantage value in our algorithm (Equation 6) to $A = r(h, \omega, u) + V(h') - V(h)$.

The experimental results displayed in Table 6 demonstrate that persona integration is achieved through three key mechanisms: (1) Behavior Cloning initializes the policy model by learning personalized patterns from offline datasets; (2) Reinforcement learning further refines personalization through persona-specific reward signals; and (3) Modeling user profile information as the unobservable state variable in the POMDP framework effectively incorporates persona features into the advantage function, ultimately enhancing the policy model's capability to generate personalized responses.

**Case Study** We present examples that illustrate the dynamic evolution of user personas across both the offline dataset and the online simulator. These cases validate the rationale behind our algorithm's approach to modeling interaction for alignment.

• **Offline dataset:** In offline datasets such as ALOE, each interaction turn only reflects a subset of (demonstrating in Table 8) the full persona information (referring to Table 7), and the persona details exhibited vary across different rounds. Therefore, this does not constitute a static persona but rather one that evolves dynamically throughout the interaction. Here is an instance from the ALOE dataset:

● **Online Simulation:** This variability, observed through LLM-based persona extraction from sampled dialogue turns in our online experiment, also serves to affirm the dynamic nature of user personas and the operational accuracy of our simulation framework. Specifically, in the implementation of the online setting, the LLM is provided with the complete user profile information at the beginning. However, through carefully designed prompts (referring to **User-Aware Prompt** in Section D), we enforce an incremental disclosure mechanism, ensuring that the simulator progressively reveals only a subset of profile details per interaction. This approach facilitates dynamic evaluation in multi-turn dialogues, where the policy model must adapt to evolving user context while preventing information overload in any single turn.

Table 9: The example for Online Simulation interaction process.

| Turn | User | Assistant | Persona ($\omega$ in Section 2) |
|---|---|---|---|
| 1 | I totally agree! Today, I rode through the local park and then took a loop around the lake. The scenery was breathtaking, and the weather was perfect. What about you? Do you have a favorite route or a special place you like to ride? | I love riding along the river trail! There's something magical about being near the water and watching the world go by. Plus, I always feel inspired to write about the scenery—do you ever bring your bike to local events or just enjoy riding solo? | enjoys cycling, enthusiastic, full of energy and passion |
| 2 | The river trail sounds amazing! I can totally see why it would be so inspiring. I usually ride solo, but I do bring my bike to school events sometimes. It's a fun way to get there, and it's great for the environment too. Do you ever ride with friends or join any cycling groups? | riding solo, it's awesome that you combine your passion with environmental impact! I enjoy riding with friends too; it's a great way to bond while sharing the love for cycling. Have you met anyone interesting through your bike rides lately? | enjoys cycling,often participates in school events,independent, capable of working alone,enthusiastic, full of energy and passion |
| 3 | I haven't met anyone new recently, but I did have a great chat with a fellow cyclist at the park last week. We talked about our favorite routes and shared some tips. It's always fun to connect with others who share the same interests. How about you? Have you met anyone interesting on your rides? | I recently spoke with a local bike enthusiast who had incredible stories about hidden trails and secret spots! It's amazing how much knowledge cyclists can share, and it sparks new adventures. Do you have any favorite spots you've discovered through your rides? | enjoys cycling,empathetic, understanding and sharing others feelings,articulate, expressing ideas clearly |

## F  USER SIMULATOR EVALUATION

Regarding the quality assessment of the simulator, we employ the offline dataset ALOE as a benchmark. Based on the dialogue history, we compare the quality of user queries generated by the simulator against those provided in ALOE. The evaluation is conducted at two levels. First, we examine whether the simulator-generated queries closely approximate the original user queries in ALOE. This comparison is performed using the Qwen3-235B-A22B-Instruct model, with a primary focus on assessing the alignment in terms of current user persona.

It is evident that approximately 76% of dialogue turns (20884 over 27480) align with user queries from the offline dataset in terms of user persona information, thereby substantiating the feasibility and rationality of the user simulator's design. The following example serves to vividly illustrate the alignment between simulator-generated queries and those in the offline dataset.

Table 10: A case study for comparison between ALOE and simulator generation.

| Persona | She is a 34-year-old freelance graphic designer who enjoys hiking, yoga, and painting. She is vegan, spicy-food lover and loves experimenting with new recipes. She loves travel around the world, especially enjoys natural scenery. She lives in a small apartment in the city with two cats named Pixel and Scribble. She is independent, capable of working alone. She is empathetic, understanding and sharing others feelings. She is methodical, working in an orderly way. She is enthusiastic, full of energy and passion. She is articulate, expressing ideas clearly. She is versatile, adapting to different roles. She is reliable, consistently dependable. She is reflective, thinking deeply about things. |
|---|---|
| History | User: Hey there! Hope you're doing well. I just got back from a hike this weekend, and the views were absolutely breathtaking! Have you been on any outdoor adventures lately? Assistant: Hey! That sounds amazing—there's nothing quite like a good hike. I haven't been on any adventures recently, but I love hearing about them! What was the highlight of your hike? |
| ALOE Query | The highlight was reaching the summit and finding a hidden waterfall. The sound of the water was so calming, and I took a moment to just soak it all in. Nature has a way of refreshing the soul, doesn't it? |
| Simulator Query | Oh, the highlight was definitely reaching the summit just as the sun was breaking through the morning fog—everything felt so still and golden. There's something about being up high with the world waking up below you. I brought my sketchbook and did a quick watercolor of the view. Makes me want to plan a trip somewhere with even bigger trails—maybe the Pacific Northwest next? |
| Aligned | Yes |
| Reason | The candidate query aligns with the user's persona by reflecting her love for nature, hiking, art, travel, and her cats, while expanding naturally on the summit experience mentioned in the actual query. |

Second, we evaluate whether the simulator-generated queries exhibit superior quality compared to those in ALOE (with the win-rate being 0.4099).

## G   LIMITATIONS

In this paper, we propose PAMDP, a dual-critic reinforcement learning framework designed to address the challenge of "Interact to Persona Alignment". Through rigorous theoretical analysis and comprehensive offline and online experimentation, we empirically validate the efficacy of our proposed approach.

Current method necessitates initializing the Actor network via behavior cloning—a prevalent practice in reinforcement learning—to constrain the exploration space and expedite convergence to near-optimal policies. However, we intend to further investigate the feasibility of eliminating this initialization paradigm, thereby streamlining the PAMDP framework for enhanced simplicity and efficiency

## H   LLM USAGE CLARIFICATION

Throughout the paper, the use of LLMs is solely restricted to the polishing of textual elements, such as lexical or phrasal substitutions, and does not extend beyond this scope.

