# OpenReview forum: "PAMDP: Interact to Persona Alignment via a Partially Observable Markov Decision Process"
_ICLR.cc/2026/Conference — ICLR 2026 Poster_

### Official Review · Reviewer_tcJ4 · 2025-10-25

**Soundness:** 3
**Presentation:** 3
**Contribution:** 2
**Rating:** 6
**Confidence:** 3

**Summary:**

This paper extends the definition of Partially Observed Markov Decision Process (POMDP) to dialogue settings. It then introduces an actor-critic algorithm to learn policy in this setting. The experiments on two datasets show that the proposed method can improve performance compared to simple baselines such as chain of thought and behavior cloning.

**Strengths:**

1. The presentation of the paper is clear.
2. The proposed method shows performance improvement on two datasets.

**Weaknesses:**

1. The proposed method relies on the quality of user simulator. If user simulator cannot well simulate real world settings which is often the case then the trained policy may work well with user simulator but may not work well under real settings.
2. It will be better if other aspects of dialogues such as informativeness and helpfulness can be evaluated.
3. The proposed method seems just a simple mapping of notations from POMDP. It seems not very novel or different.
4. More advanced RL-based dialogue improvement framework can be compared as baselines, such as [1].

References:

[1] EPO: Explicit Policy Optimization for Strategic Reasoning in LLMs via Reinforcement Learning

**Questions:**

How different is the new PAMDP, or it is just POMDP with different notations mapped to specific concepts in the dialogue setting?

---

> ### Author Response · Authors · 2025-11-20
> **Discussions on the reviews**
>
> We extend our sincere gratitude to the reviewer for the valuable insights.
>
> ## Weakness 1: The quality of user simulator
>
> Thank you for your insightful review. While user simulation is indeed designed to emulate real interaction scenarios, a discernible gap remains between simulated and authentic user behaviors. To address this limitation, we have incorporated dynamically evolving and progressively enriched user profiles in our design. As demonstrated in Table 9 (Line 928), the persona information effectively captures this evolutionary characteristic.
>
> Furthermore, our ongoing research focuses on enhancing the simulation fidelity by leveraging historical user interaction data and employing causal graph analysis to model user decision-making processes. This approach enables LLMs to generate more accurate user responses grounded in simulated cognitive reasoning. Specifically, the user's decision at time step *t*, which is conceptualized as a node value and related edge connection information within the graph, is transformed into an autoregressive function. This function models it as a causal inference outcome predicated on the entire sequence of information from the initial state to time step *t-1*.
>
> $$ p(\mathcal{E}_U, \mathcal{X}_U|\mathcal{H}, \mathcal{C}, \mathcal{E}_I, \mathcal{X}_A) = p(e_U^1,x_U^1,e_A^1,x_A^1)|\mathcal{C}) \prod_2^T\mathcal{F}_t$$
>
> $$\text{s.t.} \mathcal{F}_t = p(e_U^t,x_U^t|h^{t-1},\mathcal{C},e_U^{t-1},x_U^{t-1},e_A^{t-1},x_A^{t-1})$$
>
> where $h^{t-1}$ is the dialogue history, $\mathcal{C}$ is the user persona information, $e_U^t$ and $x_U^t$ are user decision information of current step (the node and edge are clustered by pre-defined decision texts.) Validated on a child dialogue dataset, our causal graph-enhanced LLM framework demonstrates a clear superiority over conventional prompt-based LLMs, yielding an improvement exceeding 10% to mimic the child.
>
> We acknowledge that the user simulator is indeed a critical component. However, the primary focus of this paper is to model the dynamic evolution of personalized information during interaction, a capability we have preliminarily validated through experiments on both offline datasets and an online simulator. In subsequent work, we plan to employ a more sophisticated and realistic user simulator to further advance the development of fully personalized interaction algorithms.
>
> ## Weakness 2： Other aspects of dialogues
>
> Thank you for your valuable suggestions. We have formally proposed the adoption of information gain to quantify persona information during the interaction process, as expressed in the following formulation. In subsequent work, we will incorporate corresponding experiments to better model the dynamic evolution of persona throughout the interaction.
>
> $I(\omega|s,u) = H(\omega|s_{t}) - H(\omega|s_{t},u_{t})=\mathbf{E}_{\omega \sim p(.|s_t)}[\mathbf{KL(p(\omega|s_t)||p(\omega|s_t,u_t))}]$
>
> ## Weakness 3: VS. POMDP
>
> In fact, our proposed method, i.e., PAMDP, specifically addresses the "interact to align" problem. We first provide a detailed formulation of this problem and conduct a rigorous modeling from the perspective of POMDP, where user profile information is treated as the unobservable state variable. To the best of our knowledge, this represents the first solution to multi-turn dialogue personalization from a POMDP perspective, which constitutes one of the key highlights and distinctive features of our approach. Subsequently, we derive a dual-critic formulation from the Bellman equation to solve this problem. This formulation differs from traditional actor-critic algorithms and is more inherently aligned with the problem's unique characteristics. Therefore, our primary contributions lie in both the problem definition and the innovative solution framework for "interact to align," with the dual-critic algorithm serving as an additional technical contribution. Collectively, these elements form the core of our work.
>
> ## Weakness 4: RL baselines
>
> We sincerely appreciate your suggestions. In our subsequent work, we will conduct a comprehensive comparison with several latest reinforcement learning approaches to more rigorously validate the effectiveness of our algorithm.

---

> ### Comment · Reviewer_tcJ4 · 2025-11-20
>
> Thank you for your response! However, my concerns are unaddressed. For the user simulator, it will be better if there are experiments demonstrating that the user simulator is working well in real world environment (e.g., compare the quality of the simulated of user data with real user data), instead of demonstrating that another user simulator will be available for future work. Even the function for this new user simulator is confusing: this is not an optimization problem why there is "s.j." ?
>
> For other aspects of the dialogue quality and RL baselines, it will be better if there are experiments supporting the claims, instead of saying that these aspects will be evaluated for future work.
>
> For the definition of POMDP, you also acknowledge that "this represents the first solution to multi-turn dialogue personalization from a POMDP perspective", this reinforces my concern that the proposed PAMDP is just a simple application of POMDP to multi-turn dialogue setting (this is not very novel, many existing multi-turn dialogue researches are based on this formulation)
>
> I understand and really appreciate the effort in this rebuttal, and I really want to extend my sincere gratitude for your effort, but I regret to say that this rebuttal does not address the concerns.

---

> > ### Author Response · Authors · 2025-11-27
> >
> > We appreciate the reviewer's additional comments. In response, we have conducted further validation studies to address these concerns.
> >
> > ## The quality of user simulator
> >
> > Regarding the quality assessment of the simulator, we employ the offline dataset ALOE as a benchmark. Based on the dialogue history, we compare the quality of user queries generated by the simulator against those provided in ALOE. The evaluation is conducted at two levels. First, we examine whether the simulator-generated queries closely approximate the original user queries in ALOE. This comparison is performed using the Qwen3-235B-A22B-Instruct model, with a primary focus on assessing the alignment in terms of current user persona.
> >
> > |Turns|Aligned Turns|Aligned Ratio|
> > | ----| ---- | ---- |
> > |27480|20884|75.997\%|
> >
> > It is evident that approximately 76% of dialogue turns align with user queries from the offline dataset in terms of user persona information, thereby substantiating the feasibility and rationality of the user simulator's design. The following example serves to vividly illustrate the alignment between simulator-generated queries and those in the offline dataset.
> >
> > |Key|Value|
> > |--|--|
> > |persona|She is a 34-year-old freelance graphic designer who enjoys hiking, yoga, and painting. She is vegan, spicy-food lover and loves experimenting with new recipes. She loves travel around the world, especially enjoys natural scenery. She lives in a small apartment in the city with two cats named Pixel and Scribble. She is independent, capable of working alone. She is empathetic, understanding and sharing others feelings. She is methodical, working in an orderly way. She is enthusiastic, full of energy and passion. She is articulate, expressing ideas clearly. She is versatile, adapting to different roles. She is reliable, consistently dependable. She is reflective, thinking deeply about things.|
> > |history|User: Hey there! Hope you're doing well. I just got back from a hike this weekend, and the views were absolutely breathtaking! Have you been on any outdoor adventures lately?  Assistant: Hey! That sounds amazing—there's nothing quite like a good hike. I haven’t been on any adventures recently, but I love hearing about them! What was the highlight of your hike?|
> > |ALOE query|The highlight was reaching the summit and finding a hidden waterfall. The sound of the water was so calming, and I took a moment to just soak it all in. Nature has a way of refreshing the soul, doesn’t it?|
> > |Simulator query|Oh, the highlight was definitely reaching the summit just as the sun was breaking through the morning fog—everything felt so still and golden. There’s something about being up high with the world waking up below you. I brought my sketchbook and did a quick watercolor of the view. Makes me want to plan a trip somewhere with even bigger trails—maybe the Pacific Northwest next?|
> > |Aligned|YES|
> > |Reason|The candidate query aligns with the user's persona by reflecting her love for nature, hiking, art, travel, and her cats, while expanding naturally on the summit experience mentioned in the actual query.|
> >
> > Second, we evaluate whether the simulator-generated queries exhibit superior quality compared to those in ALOE.
> >
> > ||win|loss|tie|win-rate|
> > | ----| ---- | ---- | ---- | ---- |
> > |Simulator. VS. ALOE |19166|7901|413|0.4099|
> >
> > It can be observed that the simulator-generated queries are qualitatively superior, in some respects, to the original offline data. This finding, in turn, validates the viability of the simulator.

---

> > ### Author Response · Authors · 2025-11-27
> >
> > ## RL baseline
> >
> > To establish a baseline for comparison with EPO, we processed the ALOE dataset as follows. First, LLM is employed to generate a response strategy description (e.g., "Affirmative Validation and Emotional Reflection") for each dialogue turn, conditioned on the dialogue history and persona information. The score in EPO is consistent with the per-turn reward in the ALOE dataset, with normalized to the [0, 1] range. Subsequently, we train the EPO policy model using the Llama-3B. At inference time, a response strategy is first generated based on the current dialogue history and persona, which then guides LLM to produce the final response. This output is ultimately compared against that of our end-to-end PAMDP algorithm.
> >
> > ||win|loss|tie|win-rate|
> > | ----| ---- | ---- | ---- | ---- |
> > |Ours|2255|1567|8|0.1796|
> > |EPO|2237|1580|13|0.1715|
> > |Ours. VS. EPO|1912|1911|7|0.0002|
> >
> > It is evident that the performance gain of our method is less pronounced compared to EPO. Fundamentally, EPO relies on explicit strategy generation, which then serves as a prompt to guide the LLM in response generation. In contrast, our proposed PAMDP implicitly embeds the policy into the LLM's parameters through partial information. While EPO offers greater interpretability, it requires additional annotations for strategy description. Our approach, by comparison, presents a more direct and end-to-end trainable paradigm.
> >
> > ## VS. POMDP
> >
> > At a high level, it is apt to view our approach as being grounded in the POMDP framework, which we have specifically adapted and refined for the persona-based dialogue setting.
> >
> > However, it is important to clarify that ***POMDP constitutes a broad conceptual framework and a family of algorithms***, rather than ***a single, specific implementation***. Numerous studies, including those cited in our related work, represent algorithmic innovations within this overarching POMDP paradigm.
> >
> > This is analogous to stating that a study is foundationally rooted in Reinforcement Learning, while its substantive contribution, in fact, lies in specific advancements and refinements of a concrete RL algorithm.
> >
> > Our substantive contribution, therefore, lies in being the first to formally model the "interact-to-persona" problem using the POMDP formalism. Crucially, we have further tailored the traditional POMDP structure to better capture the nuances of persona interaction and have theoretically proposed and validated a dual-critic design that is more suited to this problem.
> >
> > We hope this clarification allows you to recognize the novelty of our work from this perspective.

---

> > > ### Comment · Reviewer_tcJ4 · 2025-11-27
> > >
> > > Thanks for your comment which addresses my concern. I will keep my positive rating.

---

### Official Review · Reviewer_6rqS · 2025-10-31

**Soundness:** 3
**Presentation:** 3
**Contribution:** 3
**Rating:** 6
**Confidence:** 4

**Summary:**

This paper deals with the “Interact to Persona Alignment” problem. The authors conceptualize the issue as a Persona Consistency Markov Decision Process (PAMDP) and build a dual-critic reinforcement learning framework. They also evaluate their approach on both offline datasets and the online datasets with the user simulator, ultimately demonstrating its effectiveness.

**Strengths:**

1. I find the concept transformation method proposed in this paper highly compelling. LLMs often struggle to adapt to user preferences in real-world applications, making personalization a challenging task. This paper addresses the problem by reformulating the Persona Alignment problem as a Markov Decision Process (PAMDP) and introduces a dual-critic reinforcement learning framework to enable a more streamlined and effective personalization experience.
2. The proposed approach is notably simple (in a positive sense), intuitive, and demonstrates strong practical effectiveness.

**Weaknesses:**

1. The implementation details, especially in the experimental section, remain insufficient. Specifically, the mapping between the mathematical notations in the formulas and their actual representations in the model is unclear.
2. The use of the ALOE benchmark in offline experiments raises some concerns. The evaluation metrics differ from those in the original ALOE paper, and the reward model employed appears questionable. It is unclear whether it aligns with the original reward function.
3. In online experiments, the evaluation relies heavily on the user simulator. However, the design and reliability of the user simulator are not described in sufficient detail. It remains unclear how the simulator is built and how its quality is ensured.

**Questions:**

1. While the formalization based on POMDP is elegant, many of the core components lack clear implementation-level descriptions. For instance, what exactly are the hidden state H and the context w in the actual system? How are these represented in practice? Furthermore, why is a double-critic evaluation necessary?

2. For evaluation, you compare your method against baseline models using win rate, but the original ALOE paper provides its own evaluation metrics. Why not use those as well? Additionally, the proposed method is not directly compared to other personalized methods like BE. This makes it difficult to clearly identify the advantages of your method. A more comprehensive comparison would help clarify its empirical strengths.

3. The proposed method does not seem to construct an explicit user profile on the assistant side for reward computation. Instead, it relies on the quality of the final responses as a reward signal. How can we ensure that the user-related information embedded in the dialogue has a positive effect on model optimization? It is possible that the expected user profile may not be accurate, yet the resulting output is still good—how is this handled in your framework?

4. Regarding the dynamic evaluation in multi-turn dialogues, have you considered the impact of the user simulator’s quality on the results? How is the user simulator initialized based on the user profile? More detailed explanations would help justify the reliability of the online evaluation setting.

5. The citation format should be revised, and the meanings of each symbol should be clearly stated. In addition, the comparative experiments need to be added to the main content instead of the appendix, and the main content should illustrate more about your work.

---

> ### Author Response · Authors · 2025-11-20
> **Discussions on the reviews**
>
> We extend our sincere gratitude to the reviewer for the valuable insights.
>
> ## Weakness 1: implementation details
>
> The reinforcement learning framework comprises two integral components: the actor and the critic.
>
> The actor processes partial state information—specifically, the dialogue history $h$—and generates a user response $u$. In implementation, an LLM is employed: the historical text is input to the model, and the hidden state representation extracted (prior to the lm-head decoding layer) is projected via a transformation matrix into a low-dimensional continuous space, constituting the actual action $a$ for reinforcement learning. This low-dimensional action space facilitates efficient policy optimization. Subsequently, $a$ is mapped back to the original hidden state space and decoded to produce the response text $u$. This procedure corresponds to Equations (10) and (11).
>
> The critic component consists of two modules: partial information and full information. The partial information module takes the dialogue history $h$ as input, whereas the full information module receives the concatenated text of the dialogue history and the personalized information $\omega$. Technically, the respective text inputs—either $h$ alone or $h$ augmented with $\omega$—are fed into the LLM. The hidden state representation obtained before lm-head decoding is then projected to a scalar value, representing the estimated value of the current state, Equation (13).
>
> |Notation in formula|Representation in model|Description|
> | ---- | ---- | ---- |
> |$h$|$h$|the dialogue history; the observable state in RL|
> |$\omega$|$\omega$|the user profile or preference information; the unobservable state in RL|
> |$u$|$F(q_{\theta})$|the policy or assistant response; the action in RL|
> |$V(h)$|$V_{\phi}(h)$|the partial state value with history as the input|
> |$V(h,\omega)$|$V_{\xi}(h,\omega)$|the full state value with both history and user information as the input|
>
> ## Weakness 2: ALOE metrics
>
> In the ALOE paper, the evaluation metric employs an LLM to score responses on a scale of 1 to 5. However, this approach is inherently a generative paradigm, and its reliability heavily depends on the LLM's capabilities. Generally, we consider discriminative tasks to be simpler than generative paradigm and more likely to yield reliable results. Therefore, we have refined the metric by using the LLM to compare the relative quality of the generated response against a baseline response. This adjustment makes the task easier for the LLM, thereby enhancing the credibility of the results.
>
> Furthermore, we collected samples of comparative results from LLM and conducted human evaluations to assess evaluation plausibility. The results indicate that over 92% of the generated responses align with human judgment criteria, thereby validating the feasibility and reliability of the adopted evaluation metric.
>
> ## Weakness 3: simulator
>
> We provide the detailed prompt design for the simulator in Appendix D (line 736), where an LLM is employed to generate simulated user queries. The primary objective is to emulate the progressive and dynamic evolution of user persona information as realistically as possible. Table 9 (line 928) presents an example drawn from our online experiments, which illustrates that the persona information indeed evolves dynamically throughout the dialogue. This observation substantiates the validity and rational design of our proposed simulator.
>
> ## Question 1: double-critic evaluation
>
> In computing the advantage value, it is generally defined as $A=Q-V$, where
>  $Q$ represents the action-value function and $V$ denotes the state-value function. Under the PAMDP framework,
>  $Q=Q(h, \omega, u)$ and $V=V(h, \omega)$, as shown in Equation 5. Here,  $Q(h, \omega, u)$ can be expressed as the sum of the immediate reward $r(h,\omega,u)$ and the expected value of future states $\sum p(h'|h,\omega,u)V(h')$. In practical training, the state transition probability $p(h'|h,\omega,u)$ is typically omitted, and the next state value $V(h')$ is directly sampled to approximate the expectation over future states. Thus, the formulation should effectively use $V(h')$  for estimation.
>
> Alternatively, the expected value of future states in $Q(h,\omega,u)$ can be further decomposed to $\sum p(h'|h,\omega,u)p(\omega '|h')V(h', \omega ')$. However, if we directly approximate the expected future state value using $V$
>  alone, this necessitates disregarding both the state transition dynamics $p(h'|h,\omega,u)$
>  and the conditional probability of the unobserved variable $p(\omega '|h')$. Consequently, this approximation introduces a larger estimation error compared to using $V(h')$.

---

> ### Author Response · Authors · 2025-11-21
> **Discussions on the reviews**
>
> ## Question 2: see in weakness 2
>
> ## Question 3: user profile
>
> We incorporate two methodological safeguards to ensure the effective integration of persona information.
>
> First, within the reward structure, only responses consistent with the user’s current persona information are assigned high rewards, thus steering the model—through reinforcement learning—toward reward-maximizing behavior.
>
> Second, the loss function includes a KL divergence term that constrains the policy model’s output distribution under partial information to remain close to that of the base model under full information. This formulation encourages the policy model to internalize and leverage the current persona information at each training step. Formally, the metric is defined as:
>
> $\delta = KL(q_{\theta}(a|h)||q_{b}(a|h,\omega))= \sum (q_{\theta}(a|h) log(\frac{q_{\theta}(a|h)}{q_{b}(a|h,\omega)})) =  \sum (q_{\theta}(a|h) [log(\frac{q_{\theta}(a|h)}{q_{b}(a|h)}) + log(\frac{q_{b}(a|h)}{q_{b}(a|h,\omega)}) ])$.
>
> Since the base model remains stationary, the value of $log(\frac{q_{b}(a|h)}{q_{b}(a|h,\omega)})$ is constant. This allows us to omit $log(\frac{q_{b}(a|h)}{q_{b}(a|h,\omega)})$ and utilize the KL divergence between the action distribution of the trained model (under partial information) and that of the base model as our metric.
>
>
>
> ## Question 4: simulator’s quality
>
> In the implementation of the user simulator, the LLM is provided with the complete user profile information at the beginning. However, through carefully designed prompts (Line 777: User-Aware Prompt, Requirement 2: "You should only reveal partial information about your profile in each round of conversation instead of disclosing all the provided information at once"), we enforce an incremental disclosure mechanism, ensuring that the simulator progressively reveals only a subset of profile details per interaction. This approach facilitates dynamic evaluation in multi-turn dialogues, where the policy model must adapt to evolving user context while preventing information overload in any single turn.
>
> ## Question 5:
>
> We sincerely appreciate your valuable suggestions and will proceed to further refine the presentation of our paper.

---

### Official Review · Reviewer_wAWp · 2025-11-01

**Soundness:** 3
**Presentation:** 3
**Contribution:** 3
**Rating:** 4
**Confidence:** 4

**Summary:**

The paper formulates “Interact-to-Persona Alignment” for LLMs as a partially observable control problem, where the user’s evolving persona is a hidden state variable. It introduces the Persona Alignment MDP (PAMDP) and derives a Bellman equation specialized to this setting, a dual-critic advantage estimator, and shows the estimator is unbiased relative to the standard POMDP advantage. The actor produces a continuous latent action that steers the base LLM’s generation; critics estimate values with and without access to the hidden persona during training. Experiments span offline datasets (ALOE, PrefEval) and an LLM-based online simulator, reporting improvements over prompt/PEFT/FPFT/CoT and asymmetric-A2C baselines.

**Strengths:**

Treating persona alignment as a partially observable control problem pins down what is observed (dialogue history) vs. hidden (persona) and yields a proper Bellman structure. This makes the problem analyzable and comparable to established POMDP methods, rather than staying at the level of ad-hoc prompting.

Using one critic with history only and another with access to persona during training gives an estimator that the authors argue is unbiased relative to the true POMDP advantage, unlike standard asymmetric A2C. The actor outputs a low-dimensional latent that steers the base LLM, with KL regularization to a behavior prior (from BC).

The paper’s Theorem 3 contrasts its estimator against asymmetric A2C (UAAC/DCRL-style) and argues lower bias. Methodologically it also situates the online study alongside PPDPP’s prompt-programmed dialog planning. Relative to recent personalization datasets (ALOE, PrefEval), the focus here is on treating persona as an unobserved variable updated through interaction.

**Weaknesses:**

The results rely on LLM-as-judge and LLM-generated rewards. Both the reward used during RL and the offline/online evaluation are produced by the same family model (Qwen2.5-72B-Instruct). The same family of models is involved in generating rewards and judging outputs. No human evaluation is reported.

Cumulative returns being negative makes it unclear how “good” a policy is on an absolute scale. In the online simulator (max 6 turns), all methods—including yours—show negative cumulative returns throughout; your method is best but still ends at about −2.5. Without a calibrated scale or a normalized metric, it is difficult to infer practical significance from “less negative” numbers. Please normalize rewards (e.g., to [0,1] or standardized z-scores), report success@k and step-wise human ratings, and include qualitative trace examples that map scores to perceived usefulness. Also justify the fixed 6-turn horizon and show sensitivity to longer dialogs.

The online environment stitches together three LLM-prompted modules—Profile-Infer, User Simulator (which selectively reveals parts of the profile each turn), and a Reward Generator that mirrors the offline reward design. While useful for ablations, this pipeline may not reflect real users’ inconsistent, messy disclosures, and it inherits the same judge/reward biases. Please release the simulator prompts/code for scrutiny, stress-test with randomized persona noise and non-staged disclosures, and, if possible, run a small study with consented human participants to verify transfer.

Theorem 3 shows that the dual-critic advantage estimator is unbiased in expectation given access to the true hidden persona ω; in practice, the full-information critic uses ω while the actor never sees ω, and the system relies on LLM-inferred profiles or dataset metadata that can be noisy or partially wrong. That mismatch (and the function-approximation error in both critics) could re-introduce bias/variance. Please quantify sensitivity to persona noise and critic misspecification.

**Questions:**

NA

---

> ### Author Response · Authors · 2025-11-18
> **Discussions on the reviews**
>
> We extend our sincere gratitude to the reviewer for the valuable insights.
>
> ## Weakness 1: human evaluation
>
> In addition to the Qwen model series, our evaluation in Table 5 (line 838) incorporates the DeepSeek-R1-0528-AWQ model for win-rate assessment, which similarly substantiates the efficacy of the proposed algorithm.
>
> We further provide a subset of human evaluation results to complement our analysis. Specifically, our human evaluation is conducted at two levels.
>
> First, for response quality assessment, human judges are presented with pairs of responses—one generated by our algorithm and one by a baseline model—which are randomly shuffled and anonymized. They are then asked to select the superior response. The response pairs prepared for comparison are structured into a predefined format and presented to human evaluators via documentation. The evaluators are instructed to annotate the "human" field with a numerical code: '0' indicating Response 1 is superior, '1' indicating Response 2 is superior, and '-1' indicating the responses are of comparable quality (a tie). (LLM result is llama-8b ALOE in table 2)
>
> ### Template
> |id|input|history|persona|response 1| response 2|human|
> ### Result
> ||win|loss|tie|win-rate|
> | ----| ---- | ---- |---- |---- |
> |HUMAN|2354|1208|268|0.2992|
>
> ### Cases
> ||input|persona|response 1|response 2|
> | ----| ---- | ---- | ---- | ---- |
> |0|I’ve spotted a few around the garden! They really do brighten up the day. I love watching them play among the flowers.|He is an avid bird watcher. He is curious, always eager to learn something new.|That sounds absolutely delightful! Orioles are known for their playful behavior and beautiful songs. Watching them flutter among the flowers must be a real treat. Do you have any particular favorite spots in your garden where they seem to frequent?|That sounds amazing! Those little flutters of color must make gardening even more enjoyable. Do they have a favorite spot in your garden?|
> |1|Absolutely! Racing against a mythical creature would add so much tension and excitement. It’s a great way to explore perseverance and overcoming fears, too.|He is a graphic novelist. He enjoys cycling and goes on long bike rides.He is creative, full of innovative ideas.|Exactly! Racing against a mythical creature can symbolize overcoming personal fears and pushing through challenges. This not only adds tension but also provides a deep narrative layer about perseverance and growth. You could even use the mythical creature as a metaphor for the character’s own inner struggles or insecurities.\n\nHere are a few potential plot points to consider.|I love that angle! The race could symbolize overcoming personal challenges, like the fears that keep many cyclists from pushing their limits. It’s a great opportunity to show character growth through determination and bravery!  You could definitely highlight those themes while keeping the magical elements thrilling!|
> |-1|Thanks! I’m excited to share the results with you. It’s always fun to capture those special moments!|She is a photographer who enjoys cycling. She has a dog named Daisy. She is working on a project about local scenery. She is excited to photograph wildflowers and seeks interesting tree bark textures.She is dedicated, fully committing to her projects. She is adventurous, always seeking new experiences. She is articulate, easily expressing her thoughts.|That’s awesome! I’m looking forward to seeing your creativity come to life in those shots. Capturing those special moments really does make all the hard work worth it!|I can't wait to see your results! Capturing those special moments is always rewarding. Share them when you're ready, and I'll be here to admire your work. Happy shooting!|
>
>
> Second, we sample 500 instances from the LLM evaluation outcomes and task human judges with determining whether they agree with the reasoning provided by the LLM evaluator. We observed a 92.4% alignment between human judgments and the reasoning provided by the LLM evaluation. The results indicate a strong alignment between the LLMs' reasoning and human judgment standards, thereby corroborating the relative reliability of the LLM evaluation results.

---

> ### Author Response · Authors · 2025-11-18
> **Discussions on the reviews**
>
> ## Weakness 2: Cumulative returns for online evaluation
>
> In our experiments, the reward is formulated as follows: a powerful LLM (e.g., Qwen-72B) first generates a baseline response given the context. The output from our trained model is then benchmarked against this baseline. A reward of +1 is assigned if it surpasses the baseline, -1 if it is inferior, and 0.5 if it is deemed comparable. It is noteworthy that our model has only 8B parameters, rendering it inherently less capable than the 72B reference model. Consequently, the expected cumulative reward is naturally negative.
>
> However, as illustrated in Figure 4, the per-turn reward gain exhibits an escalating trend with the progression of dialogue turns. This indicates that the trained model progressively acquires more persona information, which in turn enhances the reward in subsequent interactions. Although the cumulative return remains negative, this trend signifies a relative improvement in the degree of personalization.
>
> We also report the online experimental results by normalizing the rewards to the interval [0, 1]. As shown below, the return demonstrates a progressive increase over the course of the dialogue, which aligns with our objective of incrementally enhancing personalization capabilities.
>
> |Turn|1|2|3|4|5|6|
> | ----| ---- | ---- |---- |---- |---- |---- |
> |UAAC|0.1446|0.3738|0.6636|1.0246|1.1981|1.5784|
> |DCRL|0.1836|0.4114|0.6419|0.8104|0.9701|1.4302|
> |Ours|0.2265|0.5242|0.7302|0.9469|1.1216|1.7389|
>
> ## Weakness 3: User Simulator
>
> The complete prompt for the user simulator is provided in Appendix D, line 737. Our design endeavored to emulate a realistic interaction process; however, a non-trivial gap persists between our simulation and real-world scenarios. Methodologically, our ongoing work is integrating concepts from cognitive psychology, such as the Big Five personality framework, to refine and quantify personalization with the goal of achieving a more faithful simulation.
>
> ## Weakness 4: Persona noise
>
> We quantify the persona noise by using the base model's action distribution under full information as a reference, computing the KL divergence between this and the distribution from the trained model under partial information. This approach is adopted because the direct computation of the state distribution is intractable. Therefore, the discrepancy in states is indirectly reflected by comparing their influences—namely, the resulting action distributions. Formally, the metric is defined as: $\delta = KL(q_{\theta}(a|h)||q_{b}(a|h,\omega))= \sum (q_{\theta}(a|h) log(\frac{q_{\theta}(a|h)}{q_{b}(a|h,\omega)})) =  \sum (q_{\theta}(a|h) [log(\frac{q_{\theta}(a|h)}{q_{b}(a|h)}) + log(\frac{q_{b}(a|h)}{q_{b}(a|h,\omega)}) ])$.
>
> Since the base model remains stationary, the value of $log(\frac{q_{b}(a|h)}{q_{b}(a|h,\omega)})$ is constant. This allows us to omit $log(\frac{q_{b}(a|h)}{q_{b}(a|h,\omega)})$ and utilize the KL divergence between the action distribution of the trained model (under partial information) and that of the base model as our metric.
>
> In fact, this variable has already been incorporated in Equation 12 (line 263). In other words, this KL divergence term serves a dual purpose: it prevents the trained policy from deviating excessively from the base model while also implicitly aiming to reduce the influence of persona noise.

---

### Official Review · Reviewer_5xM4 · 2025-11-03

**Soundness:** 3
**Presentation:** 3
**Contribution:** 3
**Rating:** 4
**Confidence:** 3

**Summary:**

This paper introduces PAMDP, a novel framework that models the task of aligning a large language model to an individual user's persona as a Partially Observable Markov Decision Process (POMDP). The authors argue that standard alignment techniques optimize for general preferences, failing to capture user-specific nuances that evolve during a conversation. In their PAMDP formulation, the user's dynamic profile is an unobservable variable, and the dialogue history is the observable state. They propose a dual-critic reinforcement learning algorithm to solve this, using an actor that only sees the observable history and two critics that access both observable and unobservable information during training to provide an unbiased advantage estimate. This approach, which uses a continuous latent action space, is validated on both offline datasets and an online simulator, demonstrating

**Strengths:**

- Originality: The paper's identify the "Interact to Persona Alignment" problem. It is the first work to formally model this problem as a Partially Observable Markov Decision Process (POMDP), which the authors term the Persona Alignment MDP (PAMDP).
- The PAMDP framework is well-justified. The authors formally derive the Bellman equation for their PAMDP (Theorem 1) and propose a novel dual-critic advantage estimator (Theorem 2). Crucially, they provide a mathematical proof (Theorem 3) that their estimator is unbiased, giving the method a strong theoretical foundation.
- The method is validated on both offline datasets (ALOE and PrefEval) using two different base LLMs (Qwen2.5-7B and Llama3-8B) and in an online setting with an LLM-based user simulator.

**Weaknesses:**

- The initial policy model (being initialized by behavior cloning. It implies the RL method may be only able to refine a policy that is already persona-aware and may not be able to learn a personalized policy from scratch. A experiment from scratch, or non-expert BC initialization, should be provided.
- The proposed method introduces a dual-critic architecture. This is more computationally expensive than the single-critic baselines or standard fine-tuning methods. The paper does not provide any analysis of this added overhead.
- The reward is only a scalar score without fine-grained feedback. The authors could explore using a more fine-grained rewards on multiple criteria, such as style, relevance, persona-consistency.

**Questions:**

- In Sec 4.1, the offline learning  trains on pre-collected datasets. Offline RL is hard due to the distributional shift. The paper uses BC initialization. How do your method mitigate the distributio shift? Will using a BC initaliation be circumventing this problem?
- The reward as well as the evaluation is using the same LLM. Wouldn't that be "circular" evaluation?
- I am not sure what "dynamically evolving profile" means. In Table 8&9, it seems to be only progressively revealed, it is not exactly what "evolving" means I think. I think the dynamically evolving profile should indicates that the user's preference is changed in the interaction, i.e., $p(\omega_{t+1} | \omega_t, h_t, u_t)$.
- Standard TD should be $\hat{A} \triangleq \delta(h,\omega,u)=r(h,\omega,u)+\gamma V(h^{\prime}, \omega)-V(h,\omega)$. Why do you use $V(h^{\prime})$ instead of $V(h^{\prime}, \omega)$.

---

> ### Author Response · Authors · 2025-11-18
> **Discussions on the reviews**
>
> We extend our sincere gratitude to the reviewer for the valuable insights.
>
> ## Weakness 1: Behavior Cloning
>
> The primary purpose of employing behavior cloning is to ensure that the actor network initially generates fluent and readable text outputs. Since our actor network is built upon the original LLM by adding new network structures after the hidden states, the output may otherwise lack readability and fail to be comprehensible to humans. Therefore, during the reinforcement learning phase, a sufficient number of exploration steps are required to first enable the model to produce coherent text, after which further customization of the output can be achieved.
>
> We have conducted additional experiments on the basis of llama-3b model to address your inquiry. Specifically, we first performed behavior cloning using the standard instruction dataset Alpaca, followed by reinforcement learning training with PAMDP. The results indicate that although no personalized training was involved during the behavior cloning phase, the final level of personalization achieved through reinforcement learning is comparable to that of other approaches.
>
> ||win|loss|tie|win-rate|
> | ----| ---- | ---- |---- |---- |
> |Base-BC+RL|2255|1567|8|0.1796|
> |Alpaca-BC+RL|2224|1594|13|0.1645|
>
> ||win|loss|tie|win-rate|
> | ----| ---- | ---- |---- |---- |
> |Alpaca-BC+RL.VS.Base-BC+RL|1888|1925|17|-0.0097|
> |Alpaca-BC+RL.VS.Alpaca-BC|2100|1709|21|0.1021|
>
> ## Weakness 2: Dual-critic architecture
>
> The critic network is designed with a lightweight architecture. Similar to the actor network, the LLM representation component remains frozen and does not participate in training. As a result, the number of parameters actually being updated is quite limited. Therefore, although the proposed dual-critic structure incurs higher training overhead compared to a single-critic framework, the absolute training time remains relatively short and falls within an acceptable range.
>
> ## Weakness 3: The reward
>
> We sincerely appreciate your suggestion. In our ongoing work, we are actively exploring the integration of psychological frameworks, such as the Big Five personality theory, into the reward modeling process. This approach aims to enhance the fidelity of persona simulation by aligning it more closely with real-world psychological constructs.
>
> ## Question 1: Distribution shift
>
> Consistent with conventional reinforcement learning approaches for LLMs, our training methodology constrains policy optimization within a proximate region of the original LLM's policy. To mitigate potential distribution shift, Equation 12 incorporates an additional KL divergence term into the objective function.
>
> ## Question 2: The same LLM
>
> Table 5 (line 838) presents evaluation results across different LLMs, which demonstrate the consistent advantage of our method. These findings indicate that our approach remains effective even when the underlying LLM differs from the one used for reward modeling.
>
> ## Question 3: Dynamically evolving profile
>
> In our paper, the concept of a "dynamically evolving profile" is defined as one that undergoes changes with each dialogue turn. This encompasses both the "progressively revealed" characteristics demonstrated in Tables 8 and 9, as well as arbitrary changes as you mentioned. The patterns shown in Tables 8 and 9 reflect the characteristics of the ALOE dataset, which align with real-world interactive scenarios. This is because, in practice, an assistant typically acquires increasing knowledge about user attributes through successive interactions, thereby constituting a progressively revealed profile.
>
> ## Question 4: TD
>
> In computing the advantage value, it is generally defined as $A=Q-V$, where
>  $Q$ represents the action-value function and $V$ denotes the state-value function. Under the PAMDP framework,
>  $Q=Q(h, \omega, u)$ and $V=V(h, \omega)$, as shown in Equation 5. Here,  $Q(h, \omega, u)$ can be expressed as the sum of the immediate reward $r(h,\omega,u)$ and the expected value of future states $\sum p(h'|h,\omega,u)V(h')$. In practical training, the state transition probability $p(h'|h,\omega,u)$ is typically omitted, and the next state value $V(h')$ is directly sampled to approximate the expectation over future states. Thus, the formulation should effectively use $V(h')$  for estimation.
>
> Alternatively, the expected value of future states in $Q(h,\omega,u)$ can be further decomposed to $\sum p(h'|h,\omega,u)p(\omega '|h')V(h', \omega ')$. However, if we directly approximate the expected future state value using $V$
>  alone, this necessitates disregarding both the state transition dynamics $p(h'|h,\omega,u)$
>  and the conditional probability of the unobserved variable $p(\omega '|h')$. Consequently, this approximation introduces a larger estimation error compared to using $V(h')$.

---

### Author Response · Authors · 2025-12-02

Many thanks to the chair and reviewers for the insightful comments. During the rebuttal process, we provide detailed responses to the reviewers’ concerns and supplemented our response with substantial additional validation experiments, which effectively address the issues raised. ***Although we have only received responses from Reviewer tcJ4 so far, the feedback indicates satisfaction with our clarifications and added evidence, and reflects a positive stance toward our work.*** We have also submitted the revised paper in accordance with the reviewers’ comments.

# A comprehensive summary
## 1. Experiments:
In response to each reviewer’s requests regarding experimental aspects—such as behavior cloning, human evaluation, reward normalization, simulator quality, and reinforcement learning baselines—we have meticulously conducted and incorporated supplementary experiments. These results, now integrated as substantial additions to the main body of the paper, effectively address the reviewers’ concerns and further validate the efficacy of our proposed method.

## 2. Theory:
We have also provided detailed clarifications and rigorous theoretical analyses in response to all reviewers' inquiries concerning algorithmic and theoretical aspects—including but not limited to TD error and persona noise. These substantive theoretical justifications are sufficient to fully resolve the reviewers’ concerns.

## 3. Revised Version:
In addition, following the reviewers' feedback and our corresponding rebuttal, we have updated the paper accordingly. Specifically, in Table 3, we have re-run the online evaluation using normalized reward and now report the updated results, ensuring that the return values are intuitively reasonable upon inspection. Correspondingly, the results in line 423 have also been revised to reflect this change. Furthermore, to substantiate the validity of our user simulator design, we have included a detailed explanation and analysis in Section F (beginning at line 961) of the revised paper.

In summary, we have thoroughly addressed all concerns raised by the reviewers and have substantiated the effectiveness of our approach through supplementary experiments in areas specifically highlighted by them. We are confident that our responses adequately resolve the reviewers’ questions. We respectfully request that the chair take our comprehensive rebuttal into full consideration when making a final assessment. Should our clarifications and additional evidence be deemed satisfactory, we sincerely hope for a favorable adjustment in the overall evaluation.

---

### Author Response · Authors · 2025-12-04

# Point-by-point restatements to each reviewer
## 1. Reviewer ***5xM4***
a. ***Behavior Cloning***: Empirical validation is conducted using the non-personalized Alpaca dataset for BC, which also achieves strong personalization through learning from scratch.

b. ***Computationally expensive***: The trainable parameters of our critic model are confined solely to a fully-connected layer appended to the frozen LLM. This design inherently results in minimal computational and memory overhead, thereby effectively circumventing any potential issue of prohibitive computational expense.

c. ***Fine-grained rewards***: The core contribution of our PAMDP work remains the validation of RL's effectiveness for persona alignment via interaction.  A novel Big-Five personality-based reward function that scores responses by comparing dimensional values is introduced in our onging work.

d. ***Distribution shift***: Incorporating KL divergence into the design of loss function can effectively mitigate the issue of distribution shift.

e. ***Circular evaluation***: Table 5 (line 838) presents evaluation results across different LLMs, which demonstrate the consistent advantage of our method.

f. ***Dynamically evolving profile***: Being defined as one that undergoes changes with each dialogue turn, showing in Tables 8 and 9.

g. ***TD***: We elucidate the rationale for employing a dual-critic architecture from the perspective of detailed theoretical error analysis, demonstrating its capacity to mitigate advantage estimation bias in comparison to conventional TD.

## 2. Reviewer ***wAWp***
a. ***Human evaluation***:  We conduct experiments involving both direct human scoring and human evaluation of LLM-generated judgments, which substantiates the efficacy of the proposed method.

b.***Cumulative returns***: We enhance the online experiments by implementing reward normalization, which yielded more pronounced evaluation outcomes. These updated results have been incorporated into the paper.

c.***User Simulator***: The complete design process of the simulator, including the full prompt specifications, is documented within the paper (Section D).

d. ***Persona noise***: We provide a detailed theoretical derivation to demonstrate that the KL-divergence loss effectively mitigates personalized noise.

## 3. Reviewer ***6rqS***
a. ***Implementation details***: We provide a clearly formatted notation table to elucidate the correspondence between the theoretical framework and the implemented model.

b. ***ALOE metrics***: The personalized nature of the content in our study is not adequately captured by the evaluation metrics proposed in the ALOE paper.

c. ***Simulator***: The design and quality assurance of the simulator can be inferred from the prompts detailed in Section D. Furthermore, Table 9 provides exemplars of the simulator’s output, which substantiate its operational reliability.

d. ***Double-critic***: We elucidate the rationale for employing a dual-critic architecture from the perspective of detailed theoretical error analysis, demonstrating its capacity to mitigate advantage estimation bias in comparison to conventional TD.

e. ***User profile***: 1) high-reward: Only responses consistent with the user’s current persona information are assigned high rewards; 2). KL loss: KL divergence term constrains the policy model’s output distribution under partial information to remain close to that of the base model under full information.

## 4. Reviewer ***tcJ4***
a. ***Simulator quality***: 1). Examine whether the simulator-generated queries closely approximate the original user queries in offline-data 2). Evaluate whether the simulator-generated queries exhibit superior quality compared to those in offline-data. Experiments for both evaluations verify the simulator quality.

b. ***EPO basline***: We process the offline data and conduct training using the EPO method. The results further validate the efficacy of our proposed approach.

c. ***Novelty***: POMDP constitutes a broad conceptual framework and a family of algorithms, rather than a single, specific implementation. Our primary contributions are threefold: first, we are the first to formally model the problem of interactive persona alignment within a POMDP framework. Second, we propose a novel dual-critic algorithm with theoretically-grounded unbiased estimation properties. Furthermore, the overall model architecture and its validation constitute a novel methodological approach.

---

### Meta-Review · Area_Chair_tWSu · 2026-01-13

**Summary:**

The paper proposes PAMDP, a framework that formulates the Interact to Persona Alignment challenge as a POMDP. The key idea is to treat the user's persona profile as an unobservable variable that evolves through interaction, where the paper introduce a dual-critic RL algorithm to solve this. Reviewers generally appreciated the rigorous problem definition and the theoretical soundness of the proposed estimator. Concerns primarily revolved around the novelty of applying a generic POMDP framework, the validity of the user simulator, and the need for human evaluation and fairer baseline comparisons.

**Reviewer Concerns:**

**Concerns addressed by rebuttal:**
- The reviewer tcJ4 initially questioned whether applying a standard POMDP framework constituted a significant contribution. The authors clarified that their contribution lies in the formal modeling of the specific "interact-to-persona" problem and the tailored dual-critic design. The reviewer explicitly acknowledged this, stating, "Thanks for your comment which addresses my concern. I will keep my positive rating."
- Concerns (by the reviewer 5xM4) about the computational expense of the critic and the necessity of Behavior Cloning (BC) were addressed. The authors explained that the critic only adds a lightweight MLP head to a frozen LLM (minimal overhead) and that BC is standard for constraining exploration, validating it with empirical results on non-personalized datasets.
- Reviewers wAWp and 6rqS requested validation beyond the simulator. The authors conducted human evaluations (both direct scoring and meta-evaluation of LLM judges) and provided detailed prompt specifications for the simulator, addressing the validity concerns.
- For the concern about theoretical rationale for dual-critic (reviewer 5xM4), the authors provided a detailed error analysis explaining how the dual-critic architecture mitigates advantage estimation bias compared to conventional TD learning, which appeared to satisfy the technical inquiries.


**Outstanding concerns:**
- Most major concerns were resolved. Some minor reservations regarding the circular evaluation (using LLMs to evaluate LLMs, raised by 5xM4) were defended by the authors via cross-model evaluation (Table 5), though this remains a general limitation of the field rather than a specific flaw of this paper.

**Reviewer Scores:**

- Reviewer 5xM4 (4->6): Likely to update to 6. The authors provided a comprehensive point-by-point rebuttal covering detailed questions (BC, cost, rewards, etc.).
- Reviewer wAWp (4->6): Likely to update to 6. The addition of human evaluation experiments directly addressed their primary request for stronger validation.
- Reviewer 6rqS (6): Likely to remain 6. Clarifications on notation and metrics were provided.
- Reviewer tcJ4 (6): Likely to remain 6. The reviewer explicitly stated they would "keep my positive rating" after the novelty concern was addressed.

---

### Decision · Program_Chairs · 2026-01-26

Accept (Poster)